# Brown carbon aerosol in rural Germany: sources, chemistry, and diurnal variations

*Feng Jiang[1,2*], Harald Saathoff[1*], Junwei Song[1], Hengheng Zhang[1], Linyu Gao[1], and Thomas Leisner[1,3]*

[1]Institute of Meteorology and Climate Research, Karlsruhe Institute of Technology, 76344 Eggenstein–Leopoldshafen, Germany
[2]Institute of Applied Geosciences, Working Group for Environmental Mineralogy and Environmental System Analysis, Karlsruhe Institute of Technology, 76131 Karlsruhe, Germany
[3]Institute of Environmental Physics, Heidelberg University, 69120 Heidelberg, Germany

*Correspondence to:* Feng Jiang (feng.jiang@kit.edu) and Harald Saathoff (harald.saathoff@kit.edu)

**Abstract**. Brown carbon aerosol (BrC) is one major contributor to atmospheric air pollution in Europe, especially in winter. Therefore, we studied the chemical composition, diurnal variation, and sources of BrC from 17[th] February to 16[th] March at a rural location in southwest Germany. In total, 178 potential BrC molecules (including 7 nitro aromatic compounds, NACs) were identified in the particle phase comprising on average $83 \pm 44$ ng m$^{-3}$, and 31 potential BrC (including 4 NACs) molecules were identified in the gas phase contributing on average $8.5 \pm 6.7$ ng m$^{-3}$ during the whole campaign. The 178 potential BrC molecules only accounted for $2.6 \pm 1.5\%$ of the total organic mass, but can explain $14 \pm 13\%$ of the total BrC absorption at 370 nm, assuming an average mass absorption coefficient at 370 nm (MAC$_{370}$) of 9.5 m$^2$ g$^{-1}$. A few BrC molecules dominated the total BrC absorption. In addition, diurnal variations show that gas phase BrC was higher at daytime and lower at night. It was mainly controlled by secondary formation (e.g. photooxidation) and particle-to-gas partitioning. Correspondingly, the particle phase BrC was lower at daytime and higher at nighttime. Secondary formation dominates the particle-phase BrC with $61 \pm 21\%$, while $39 \pm 21\%$ originated from biomass burning. Furthermore, the particle-phase BrC showed decreasing light absorption due to photochemical aging. This study extends the current understanding of real-time behaviors of brown carbon aerosol in the gas and particle phase at a location characteristic for the central Europe.

## 1. Introduction

The Brown Carbon (BrC) aerosol has significant impact on air quality and climate, since it absorbs the solar radiation in the near-ultraviolet and visible region (Laskin et al., 2015; Moise et al., 2015). Global simulation showed that the mean radiative forcing of BrC aerosol was -0.43 W m$^{-2}$ and 0.05 W m$^{-2}$ at the surface and at the top of the atmosphere, accounting for 15% of total radiative forcing by the absorbing aerosol (Park et al., 2010). In addition, global measurements of BrC found that the average direct radiative effect of BrC absorption accounted between 7% to 48% at the top of the atmosphere (Zeng et al., 2020).

Some typical molecules of BrC have been identified, such as nitro-aromatic compounds (NACs), imidazoles, and polycyclic aromatic hydrocarbons (PAH), etc., (Jiang et al., 2022; Wu et al., 2018; Huang et al., 2018; Liu et al., 2023). In western Europe, the concentration levels of NACs range between 1–20 ng m$^{-3}$, accounting for 0.3%–4% of total absorption of BrC at UV wavelengths (Jiang et al., 2022; Mohr et al., 2013; Teich et al., 2017). In addition, imidazoles were detected with concentrations ranging between 0.2–14 ng m$^{-3}$ in ambient aerosol samples from different environments in Europe and China (Teich et al., 2016). Furthermore, parent-PAHs and carbonyl-OPAHs accounted for on average ~1.7% of the overall absorption of methanol-soluble BrC in Urban Xi'an, Northwest China (Huang et al., 2018). Even though many studies have investigated the chemical composition of brown carbon and calculated the absorption contribution from BrC molecules, there are still many unknown brown carbon molecules to allow a quantitative assessment of their sources and atmospheric impact.

Sources of BrC can be separated as primary emissions and secondary formation. The primary sources of BrC are biomass burning and fossil fuel combustion (Andreae and Gelencser, 2006). On a global scale, a majority of BrC aerosol mass is associated with biomass burning dominating BrC absorption (Zeng et al., 2020). The major secondary sources of brown carbon are from oxidation of aromatic volatile organic compounds, such as toluene (Lin et al., 2015), naphthalene (Siemens et al., 2022), ethylbenzene (Yang et al., 2022), and indole (Montoya-Aguilera et al., 2017; Jiang et al., 2023), especially in the presence of $NO_2$.

BrC in the atmosphere can be suspended in the gas phase or particle phase. However, only a few studies have investigated the sources and chemical composition of BrC in the gas phase. For example, NACs in the gas phase were highest during the daytime at a rural site in China (Salvador et al., 2021). The major sources of NACs were from secondary formation on days without extensive biomass burning emissions, but mainly from primary emissions in biomass burning events (Salvador et al., 2021). The source of nitrophenol, a typical BrC molecule, was mainly from secondary formation overweighting losses by photolysis in polluted urban environments, Beijing (Cheng et al., 2021).

The major chromophores of BrC in the gas phase were rich in phenol- and protein-like substances in Xi'an, China,
during the summer (Chen et al., 2021). Therefore, the previous studies mainly focus on sources and chromophores of
BrC, especially NACs. However, the real-time diurnal variation and sources of BrC in the gas phase in the atmosphere
have rarely been investigated in central Europe.
Previous field studies have investigated the sources of BrC in the particle phase which are mainly from secondary
formation and primary emissions (Wang et al., 2019a; Moschos et al., 2018; Satish et al., 2017). In the central Europe,
the secondary biogenic organic aerosol (OA) contributes less BrC in summer. However, the primary and secondary
wood burning emissions dominated the BrC (Moschos et al., 2018). The primary emissions of BrC contributed more
to organic aerosol light absorption than those from secondary processes in the North China Plain, China (Wang et al.,
2019a). However, secondary sources for BrC were more important for absorption than primary ones in the Southeastern
Margin of Tibetan Plateau (Wang et al., 2019b). Loss pathways of BrC in the particle phase mainly comprise
photooxidation and photobleaching, but also dilution of BrC e.g. by rising boundary layer height influences its
concentration levels (Satish et al., 2017; Laskin et al., 2015; Moise et al., 2015). The absorption of BrC was high in
the early morning and later decreased due to the bleaching of chromophores (Wang et al., 2019a; Satish et al., 2017).
A diurnal cycle showed that secondary chromophores can be formed from photochemical oxidation after sunrise
followed by photobleaching of the chromophores under the oxidizing conditions as the day progressed (Wang et al.,
2019b). Lower BrC concentrations during noon were explained by the fact that planetary boundary layer heights were
highest during the middle of the day (Liu et al., 2023). However, also nighttime aqueous-phase chemistry can promote
the formation of secondary light absorbing compounds and the production of strongly absorbing particles (Wang et al.,
2019a). In addition, higher emissions of biomass burning BrC were observed at nighttime. Actually, the BrC in the
particle phase undergoes complex photochemical processing during the whole day. The time dependent sources and
diurnal variations of BrC in aerosol particles are still reported rarely and not well understood.
To better understand the chemical characterization, diurnal variation, and sources of BrC in central Europe, we
performed online measurements of BrC during February-March 2021 at a rural location in southwest Germany. In the
following, we will describe the experimental methods used in this study. Subsequently, the mass concentrations of BrC
in gas and particle phase will be determined. Furthermore, the contribution of BrC to light absorption in the particle
phase will be estimated. Then, the diurnal variations and sources of BrC in the gas and particle phase will be analyzed.
Finally, the atmospheric implications of our findings will be discussed.

## 2. Experimental methods

### 2.1. Measurement site

We performed particle and trace gas measurements from February 17th–March 16th 2021 at KIT Campus Nord, a rather rural area in Germany (49°05'43.1"N 8°25'45.6"E). The sampling site is located at the building number 322 of the IMK-AAF on KIT Campus Nord, as shown in Figure S1. The campus is mostly surrounded by the Hardwald forest dominated by pine trees. The sampling site is also near some villages e.g. 3–4 km east of the village "Eggenstein-Leopoldshafen", 6–7 km northeast of the village "Neureut", 3–4 km west of the village "Friedrichstal", 4–5 km northwest of the village "Stutensee", and 5–6 km southeast of the village "Linkenheim". Therefore, influences by biomass burning emissions from wood stove combustion in these residential areas during winter time can be expected (Thieringer et al., 2022). Furthermore, the city of Karlsruhe with 3000000 inhabitants is 10 km south of the measurement site. The city includes industrial areas with a coal-fired power plant "Rheinhafen" and a refinery "MIRO". Therefore, the measurement site is potentially affected by different aerosol sources.

### 2.2. Meteorological, aerosol particle, and traces gas instruments

All instruments were set up in a temperature-controlled measurement building. The samples were collected above the roof top about 8 m above ground level via stainless steel tubes and a $PM_{2.5}$ and a TSP inlet as well as FEP tubes for the VOC measurements. An overview of the instruments used and the parameters measured is given in Table S1 of the Supplement.

Temperature, relative humidity (RH), pressure, wind speed, wind direction, precipitation, and global radiation were measured by a meteorological sensor (WS700, Lufft GmbH; see Table S1) about 8 m above the ground level. The main wind directions during the campaign were southwest, northeast, and southeast, since winds were channeled by the Rhine River valley. $O_3$ and $NO_2$ were measured with standard gas monitors (Table S1). The particle number concentrations (>2.5 nm) were measured by a water-based condensation particle counter (CPC3789, TSI Inc.). $PM_{2.5}$ was measured by an optical particle counter (OPC-FIDAS 200, Palas Inc.). The particle number size distributions were measured by a nanoparticle sizer (NanoScan, TSI Inc.) ranging from 10-410 nm at a time resolution of 1 min. Black carbon (BC) concentrations were measured with aethalometers (AE33, Aerosol Magee Scientific).

## 2.3. Online FIGAERO-CIMS measurement and identifications of potential BrC molecules

The individual organic compounds in both the gas and particle phase were measured with a filter inlet for gases and aerosols coupled to a high-resolution time-of-flight chemical ionization mass spectrometer (FIGAERO-HR-ToF-CIMS, Aerodyne Research Inc. hereafter CIMS) employing iodide ($I^-$) for chemical ionization (Lopez-Hilfiker et al., 2014; Jiang et al., 2022). During the gas-phase measurement, the ambient air was sampled via a fluorinated ethylene propylene (FEP) tube of 4.5 m length (flow rate 8 L min$^{-1}$, residence time 0.9 s). At the same time, the particles were collected on a Teflon (Ploytetrafluoroethylene, PTFE) filter via s separate sampling port connected to a PM$_{2.5}$ inlet (total flow rate 16.7 L min$^{-1}$) and an 8 m long stainless-steel tube. The loading time and sampling flow of Teflon filters were 30 minutes and 4 L min$^{-1}$, respectively. At regular intervals (46 min), the gas-phase measurement was switched off and particles on the filter were desorbed by a flow of ultra-high-purity nitrogen (99.9999 %) heated from room temperature to 200 ∘C over the course of 35 min (Lopez-Hilfiker et al., 2014; Huang et al., 2019a). The resulting mass spectral signal evolutions as a function of desorption temperature are termed thermograms (Lopez-Hilfiker et al., 2014). Integration of thermograms of individual compounds yielded their signal in counts per second, which were converted to mass concentrations using an average sensitivity of 22 count s$^{-1}$ ppt$^{-1}$ (Lopez-Hilfiker et al., 2014). After the filed campaign, the calibration of 4-nitrophenol, 4-nitrocatechol, 2-methyl-4-nitropehnol, and 4-methyl-5-nitrocatechol was utilized to characterize the sensitivity factor of nitro aromatic compounds (NACs), as shown in the Supplement. The sensitivity factors of our iodide CIMS for 4-nitrophenol, 4-nitrocatechol, 2-methyl-4-nitropehnol, and 4-methyl-5-nitrocatechol were $0.80 \pm 0.44$, $0.50 \pm 0.32$, $0.96 \pm 0.52$, $0.97 \pm 0.63$, respectively (Figure S9). The average sensitivity factor of 4 NACs was $0.81 \pm 0.53$. We used this average sensitivity factor to calibrate other potential brown carbon molecules in this study. The sensitivity factor of levoglucosan was $0.40 \pm 0.14$ in this study (Figure S10). We used the sensitivity factor of $0.40 \pm 0.14$ to estimate the concentrations of molecules, which are not identified as potential BrC molecules. Please note that the sensitivity of CIMS for different organic compounds varies by a few orders of magnitude. Sensitivity uncertainties were taken into account in the calculation of the overall uncertainties of CIMS concentrations ($\pm 60\%$) following the approach by Thompson et al. (2017).

During the measurements, the mass resolution of FIGAERO-CIMS was relatively stable with about 4000 m/Δ. The interference from isomers with different vapor pressures or thermal fragmentation of larger oligomeric molecules can lead to more complex, multimodal and broader thermograms (Lopez-Hilfiker et al., 2014). The signal integration can include the different isomers or thermal fragmentation of larger oligomers. Therefore, the isomers or thermal

decomposition can lead to increase errors of estimating the organic mass concentrations. In this study, BrC molecules
were identified and partially quantified in atmospheric aerosol by FIGAERO-CIMS. Please note that the iodide CIMS
has sensitivities varying over several orders magnitude for different compounds e.g. of different oxidation states
(Lopez-Hilfiker et al., 2016). Therefore, the quantitative interpretation is limited to the small amount of compounds
for which we could do calibration with authentic standards. Keeping this in mind, it can still be meaning to a relative
comparison of the large number of high oxidized compounds assuming the same sensitivity. The raw data were
analysed by using the toolkit Tofware (v3.1.2, Tofwerk, Thun, Switzerland, and Aerodyne, Billerica) with the Igor Pro
software (v7.08, Wavemetrics, Portland, OR). Gas phase background was determined by sampling zero air (high purity
synthetic air). Particle phase backgrounds were assessed by putting an additional Teflon filter upstream of the particle
phase sampling port during the deposition (Huang et al., 2019a; Lee et al., 2018).
We observed typically about 1500 mass peaks from particles and 120 mass peaks in gases corresponding to different
oxygenated organic compounds by using FIGAERO-CIMS. Individual compounds were assigned to the mass peaks
by fitting, $C_cH_hO_oN_n$, different numbers of atoms: c carbon, h hydrogen, o oxygen, n nitrogen (Lopez-Hilfiker et al.,
2014). A double bond equivalent (DBE) can be calculated as follows (Daumit et al., 2013):
$$DBE = \frac{n-h}{2} + c + 1 \tag{1}$$
Lin et al., (2016, 2018) employed high-resolution mass spectrometry to analyze biomass burning organic aerosol. They
assigned potential brown carbon compounds according to the correlation of double bond equivalents (DBE) with the
number of carbon atoms per molecule (Figure S12). We used this method to assign 178 potential BrC molecules
(including 7 NACs) in the particle phase and 31 potential BrC molecules (including 4 NACs) in the gas phase, as
shown in Figure 1 in the corresponding mass spectra. The gas to particle phase partitioning coefficients of those semi
volatile potential brown carbon molecules which could be measured in both phases with sufficient sensitivity are listed
in table S6. A few other studies used this method also to assign more brown carbon molecules. For example, good
correlations (r = 0.9) between mass absorption efficiency at 365 nm and potential brown carbon molecules of larger
molecular weight were found by Tang et al., (2020). Xu et al., (2020) used this method to assign 149 nitrogen-
containing potential BrC chromophores at the Tibetan Plateau and we used this method to assign potential BrC
molecules in downtown Karlsruhe (Jiang et al., 2022). The potential BrC molecules we assigned according to this
method for the particle and the gas phase are listed in Tables S2 and S3.

**2.4. Particle light absorption from aethalometer measurements**

In the aethalometer AE33 (Magee Scientific), aerosol particles are continually sampled on a quartz filter and the optical attenuation is measured with time resolutions 1 minute at seven wavelengths (370, 470, 520, 590, 660, 880, and 950 nm) during this campaign. The light absorption at seven wavelengths was calculated from the measured attenuation. Attenuation is measured on two spots with different sample flows and on the reference spot without sample flow. The two loading spots with different flow are used to allow for loading effect corrections (Drinovec et al., 2015). Since our aethalometer has been used two loading spots, the loading effect was corrected by a Dual-spot loading compensation algorithm (Drinovec et al., 2015). To further address the scattering effect (Yus-Díez et al., 2021), we did comparison experiments in the Aerosol Preparation and Characterization (APC) chamber (Huang et al., 2018). Black carbon was injected into the APC chamber by using the PALAS soot generator (GfG 1000, Palas) (Saathoff et al., 2003). The APC chamber was connected to a photoacoustic spectrometer (PAS) operating at three wavelengths (405, 520, and 658 nm) (Linke et al., 2016) and an aethalometer AE33. As shown in Figure S11, for three wavelengths (370, 520, and 660 nm), the correlation slopes were 1.88, 1.94, and 1.98, respectively. The average multiple-scattering correction factor was $1.90 \pm 0.06$ in this study.

The BC mass concentration is calculated from the change in optical attenuation at 880 nm in the selected time interval using the mass absorption cross section 7.77 $m^2\,g^{-1}$ (Gundel et al., 1984), since other aerosol particles (organic aerosol or mineral) have less absorption at this wavelength and major absorption is contributed from BC alone. The attenuation mass absorption coefficients of AE33 from 370 – 880nm were 18.47, 14.54, 13.14, 11.58, 10.35, and 7.77 $m^2\,g^{-1}$, respectively. The absorption measurements by aethalometer have the filter-based lensing effect (Moschos et al. 2021). According to previous studies, the uncertainty from lensing effect for BC and BrC measurement were 8%-27% and 6%-20%, respectively (Moschos et al. 2021). We assumed an $AAE_{BC}$ value of 1.0 in this study. However, this assumption introduces an uncertainty in the estimations of BC and BrC light absorptions. According to previous studies, the $AAE_{BC}$ ranges between 0.8-1.4 (Lack and Langridge 2013). This range although maybe not fully applicable to our measurement location, potentially causes relatively large uncertainties of up 81% (at 370nm) in splitting between BrC and BC absorption (Figure S13) (Duan et al. 2024). Despite these potentially large uncertainties on absolute absorption values, we consider this method still useful. Our assumption of $AAE_{BC} = 1.0$ is reasonable for our location as based on previous measurements and it should still allow to discuss the relative evolution of BC and BrC absorption.

We assumed that the absorption from dust and other aerosol was negligible. Hence, the absorption was only
contributed from BC and BrC. Therefore, Abs(λ) can be divided in BC and BrC absorption:
$Abs = Abs_{BrC}(\lambda) + Abs_{BC}(\lambda)$ (2)
where $Abs_{BrC}(\lambda)$ is the absorption caused by BrC at the following aethalometer wavelengths, λ = 370, 470, 520, 590,
or 660 nm while $Abs_{BC}(\lambda)$ is the absorption contributed by BC at the same wavelength (Wang et al., 2019). To
determine $Abs_{BC}(\lambda)$ at each wavelength, we assumed that BC was the only absorber at λ = 880 nm, and thus the $Abs_{BC}(\lambda)$
(λ = 370, 470, 520, 590, and 660) can be extrapolated from the following equation:
$Abs_{BC}(\lambda) = Abs_{880} \times (\frac{\lambda}{880})^{-AAE_{BC}}$ (3)
where $AAE_{BC}$ represents the spectral dependence of $Abs_{BC}(\lambda)$, and a value of 1.0 was chosen for $AAE_{BC}$ based on
previous studies in Germany (Teich et al., 2017). Finally, one can obtain the $Abs_{BrC}(\lambda)$ as follows:

$Abs_{BrC}(\lambda) = Abs(\lambda) - Abs(880) \times (\frac{\lambda}{880})^{-AAE_{BC}}$ (4)
The fraction of wood burning black carbon (BCwb) was calculated by using the Aethalometer model (Sandradewi et
al., 2008a; Sandradewi et al., 2008b):
$BC_{wb} = [\frac{b_{abs}(470nm) - b_{abs}(950nm) * (\frac{470}{950})^{-\alpha ff}}{(\frac{470}{950})^{-\alpha wb} - (\frac{470}{950})^{-\alpha ff}}] / b_{abs}(950nm) * BC$ (5)
Where two pairs of Ångström exponents values were utilized to obtain BC associated with fossil fuel (BCff) and wood
burning (BCwb). One of the largest sources of uncertainty in the Aethalometer model is related to the section of αff and
αwb values (Healy et al., 2017; Zotter et al., 2017). In addition, the αff was typically in the range of ∼ 0.8 − 1.2 in
ambient air whereas αwb can vary from 1.6 to 2.2 (Saarikoski et al., 2021). However, we used the αff and αwb values
as 0.95 and 1.68 to calculate the BC source (Helin et al., 2018), since our measurement site is in a rural area and nearby
a suburban area.
**3. RESULTS AND DISCUSSION**
**3.1. Overview of the field observations**
Figures S1 and S2 give an overview of the measurement location and the meteorological parameters, traces gases,
particle concentrations, and their optical properties during the campaign. The major wind directions at KIT Campus
Nord, 3 km east of the village of Eggenstein-Leopoldshafen, were northeast and southwest (Figure S1) caused by
channeling of the wind in the Rhine valley. The average wind speeds were 1.1 ± 0.8 (average ± standard deviation) m
s$^{-1}$. Depending on meteorological conditions, local sources and regional transport had a major impact on air quality in
Leopoldshafen in summer (Shen et al., 2019). As shown in Figure S5, $O_3$ had diurnal variations with peaks at daytime
and an average of $41.3 \pm 26.2$ µg m$^{-3}$ during the campaign. In contrast, the relative humidity (RH) showed diurnal
variations with peaks at nighttime and an average of $68 \pm 16\%$ during the campaign (Figure S5). The average
temperature during the winter campaign was $6.5 \pm 5.6$ °C and slowly increased from beginning to the end of the
campaign. $NO_2$ had high concentrations at some periods e.g. from 20[th] to 23[th] February with $22 \pm 8.6$ µg m$^{-3}$ and from
2[nd] to 4[th] March with $35 \pm 14$ µg m$^{-3}$. The average $SO_2$ concentration was $0.8\pm1.0$ µg m$^{-3}$, significantly lower than the
$NO_2$ concentrations. During some Saharan dust events, the $PM_{2.5}$ and $PM_{10}$ mass concentrations were $21 \pm 6$ and $45 \pm$
$20$ µg m$^{-3}$, respectively, from 18[th] to 26[th] February and $19 \pm 6$ and $24 \pm 7$ µg m$^{-3}$, respectively, from 1[st] to 4[th] March as
indicated by red boxes in the lowest panel of Figure S2. In addition, BC showed many spikes and a good correlation
($r = 0.8$) with $NO_2$ (Figure S3). This indicates that there were many combustion events during the campaign (Figure
S3). The absorption Ångström exponents of particles between 370 and 520 nm ($AAE_{370-520}$) and $AAE_{660-950}$ had diurnal
variations with peaks at nighttime. We calculated the fraction of wood burning BC and fossil fuel BC as shown in
Figure S3 using the Aethalometer model (Sandradewi et al., 2008a). During the winter campaign, the biomass burning
BC was on average $0.61 \pm 0.0.49$ µg m$^{-3}$, mostly higher than $0.0.25 \pm 0.27$ µg m$^{-3}$ for fossil fuel BC. The $AAE_{370-520}$,
$AAE_{660-950}$, biomass burning BC, and $NO_2$ values were enhanced from 20[th] to 23[th] February and 2[nd] to 4[th] March. This
indicates that strong biomass burning (BB) events were on these days. During this winter campaign, the BrC absorption
accounted for ~40% of total absorption caused by BC and BrC at 370 nm. This points to the at least regional or seasonal
importance of BrC absorption which has an important effect on air quality and climate.
**3.2. Mass concentrations and volatility of potential brown carbon molecules**
Figure 2 shows an overview of levoglucosan concentrations, BC concentrations, absorption of brown carbon at 370
nm ($b_{brc370}$), $AAE_{370-520}$, volatility and mass concentrations of 178 potential brown carbon molecules identified in the
particle phase and 31 potential brown carbon molecules in the gas phase during the whole winter campaign. We
identified 178 potential BrC molecules according to the method developed by Lin et al., (2018) (cf. section 2.3.). The
mass of these molecules shows a good correlation ($r = 0.7 \pm 0.1$) with the absorption at 370 nm ($b_{BrC370}$) of BrC (sf.
Figure S6). This indicates that it is meaningful to extract these 178 potential BrC molecules from more than one
thousand and five hundred molecules detected by FIGAERO-CIMS based on the double bond equivalent/carbon
number ratio (DBE/C) of each molecule being higher than 0.5 and less than 0.9. The levoglucosan showed a good
correlation (r = 0.7) with BC. This is in line with the large fraction of biomass burning contributing to BC during the
winter campaign. Biomass burning BC accounted for (71 ± 40)% of total BC as we discussed above.  The 178 potential
BrC molecules detected in the particle phase correspond to an average mass concentrations of 83 ± 44 ng m$^{-3}$. In
addition, the nitro aromatic compounds (NACs) were also detected during the winter campaign. The mass
concentration of $\sum$NACs in the gas phase and particle phase were 1.9 ± 1.5 ng m$^{-3}$ and 17.5 ± 18.4 ng m$^{-3}$, respectively
(Table S4 and S5). Mohr et al. (2013) found that five BrC molecules (nitro aromatic compounds) were 20 ng m$^{-3}$
detected by CIMS during winter in Detling, United Kingdom. Jiang et al. (2022) measured an average concentration
of five BrC molecules (nitro aromatic compounds) of 1.6 ± 0.9 ng m$^{-3}$ during the winter at a kerbside in downtown
Karlsruhe, a city in southwest Germany and close to our measurement site. Therefore, the detection of the 178 potential
BrC molecules allows more complete assessment of the BrC concentrations during this winter campaign. Their
concentrations were significantly higher for biomass burning (BB) events e.g. 144 ± 41 ng m$^{-3}$ at BB event 1 and 124
± 39 ng m$^{-3}$ at BB event 2, respectively. In addition, the absorption of brown carbon at 370 nm (b$_{brc370}$) had high peaks
with ~100 Mm$^{-1}$ and the AAE$_{370-520}$ of particles increased from ~1.5 to ~2 during the BB events. The average
concentration of potential BrC in the gas phase was 8.5 ± 6.7 ng m$^{-3}$ during the winter campaign. At BB events, their
concentration can reach up to 38 ng m$^{-3}$. Therefore, biomass burning had a significant impact on optical properties of
aerosol and brown carbon concentrations. The lowermost panel of Figure 2 shows the temporal variation of the average
volatility of brown carbon molecules in the gas and particle phase. The average volatility or saturation concentration
(log$_{10}$C$_{sat}$) of potential BrC in the particle phase was with -1.1 ± 0.5 µg m$^{-3}$ lower than 0.9 ± 0.6 µg m$^{-3}$ of potential BrC
in the gas phase during the winter campaign. Organic compounds with log$_{10}$C$_{sat}$ lower than −4.5 µgm$^{-3}$, between −4.5
and −0.5 µgm$^{-3}$, between −0.5 and 2.5 µgm$^{-3}$, and between 2.5 and 6.5 µgm$^{-3}$ are termed extremely low-volatility
organic compounds (ELVOCs), low-volatility organic compounds (LVOCs), semi-volatile organic compounds
(SVOCs), and intermediate-volatility organic compounds (IVOCs), respectively (Donahue et al., 2009). Therefore,
BrC in the particle phase can be classified on average to the LVOCs and BrC in the gas phase to the SVOCs.
**3.3 Absorption contribution of nitroaromatic compounds and potential brown carbon molecules**
Black carbon dominated light absorption of aerosol particles with a contribution of 100% at 880 nm and decreasing to
73% at 370 nm. With shorter wavelengths, the brown carbon absorption contribution significantly increased
contributing 27% of total aerosol absorption at 370 nm (Figure 3a). We have no independent quantification of the total
organic aerosol mass loadings. However, we estimated the total organic mass as a fraction of 50 ± 20 % of PM$_{2.5}$ which
is a typical fraction for at the location (Song et al., 2022; Song et al., 2024). According to this assumption, the average
organic mass concentration was $4.5 \pm 3.1$ µg m$^{-3}$. The organic mass detected by FIGAERO-CIMS based on calibrated
sensitivity factors was $37 \pm 20\%$ of the estimated organic mass. This is in a similar range as observed in previous
studies (Ye et al., 2021). We calculated the light absorption of NACs by using molecular MAC$_{365}$ (Xie et al., 2017),
as shown in Table S5. Based on this, the mean light absorption of the sum of the seven NACs was calculated to be 0.2
$\pm 0.2$ Mm$^{-1}$, contributing to $2.2 \pm 2.1\%$ of total BrC absorption at 370 nm, but they only contributed $0.45 \pm 0.32\%$ of
the total organic mass.
In order to calculate the light absorption from the other 171 potential brown carbon molecules identified, we assumed
an average MAC value of 9.5 m$^2$ g$^{-1}$ at 370 nm for all BrC molecules to estimate their absorption (Jiang et al., 2022).
So far, the MAC$_{370}$ of most potential brown carbon molecules are still unknown. In addition, since the potential BrC
molecules detected by FIGAERO-CIMS could have isomers effect, we did not calibrate mass absorption coefficients
of 171 potential BrC. Despite these uncertainties, we think it is reasonable to estimate the order magnitude of the total
BrC absorption based on this assumption. Based on this assumption, we calculated the light absorption of the 171
potential brown carbon molecules identified to $0.6 \pm 0.3$ Mm$^{-1}$ at 370 nm as average for the whole winter campaign.
This is half the values Jiang et al. (2022) found as mean light absorption of 316 potential BrC molecules of $1.2 \pm 0.2$
Mm$^{-1}$ at 365 nm for downtown Karlsruhe in winter. Relative to this total organic aerosol particle mass and the measured
brown carbon absorption, the 171 potential identified brown carbon molecules and 7 NACs only accounted for $2.6 \pm$
$1.5\%$ of the total organic mass, but explain $14 \pm 13\%$ of total brown carbon absorption at 370 nm (Figure 3b and 3c).
Palm et al. (2020) found that particulate nitroaromatic compounds (BrC molecules) can explain $29 \pm 15\%$ of average
BrC light absorption at 405 nm, despite accounting for just $4 \pm 2\%$ of average OA mass in fresh wildfire plumes. Mohr
et al. (2013) found that five nitroaromatic compounds (BrC molecules) are potentially important contributors to
absorption at 370 nm measured by an aethalometer and account for $4 \pm 2\%$ of UV light absorption by brown carbon in
Detling, United Kingdom during winter. Jiang et al. (2022) determined a mean light absorption of the 316 potential
BrC molecules accounting for $32 \pm 15\%$ of methanol-soluble BrC absorption at 365nm, but only accounted for $2.5 \pm$
$0.6\%$ of the organic aerosol mass. Therefore, even small mass fractions of strongly absorbing brown carbon molecules
can dominate the brown carbon absorption.

**3.4 Diurnal variations and sources of potential BrC in the gas phase**

As shown in Figure 4a, the 31 gas-phase potential BrC (GBrC) molecules showed higher concentrations at daytime (09:00-17:00) and lower concentration between evening and early morning (18:00-08:00). Salvador et al. (2021) also found that 16 gas-phase nitro-aromatic compounds (BrC molecules) measured by FIGAERO-CIMS were higher during daytime and lower at nighttime during winter in rural China. As discussed above, strong biomass burning emission were mostly observed at evening and early morning hours. However, gas-phase BrC had no peaks during those time periods. Therefore, the primary emission from biomass burning was not a major source for GBrC at KIT Campus Nord. It seems to be mainly controlled by secondary formation (e.g. photochemical smog) or/and particle-to-gas partitioning (Salvador et al., 2021).

To demonstrate how secondary formation and partitioning control the gas-phase BrC in rural Germany, we plotted diurnal profiles of the average volatility and volatility fractions of IVOC, SVOC, and LVOC of the gas-phase BrC (Figure 4b). The LVOC of BrC increased at evenings and decreased at daytime. In contrast, the IVOC of BrC increased at daytime and reached ~17% of total $\log_{10}C^*$ (volatility) in gas-phase BrC while SVOC remained with a relative constant fraction (~60%). Furthermore, the IVOC fraction of BrC in the particle-phase was only 1.5% with a flat diurnal profile (Figure S7). The O/C ratio of gas-phase BrC also increased during daytime (Figure 4d). Therefore, the higher fraction of IVOC in the gas phase at daytime is most likely caused by secondary formation e.g. photochemical conversion/aging because of higher oxidant levels as indicated e.g. by higher concentration of ozone at same time (Figure 4c) (Saarikoski et al., 2021). Figure S8 shows that BrC in the gas phase had a moderate positive correlation (r = 0.4) with temperature. This explains why the temperature shows a similar diurnal profile as the gas-phase BrC. Therefore, particle-to-gas partitioning was also an important source for gas-phase BrC. However, our results are not consistent with previous studies where 16 BrC molecules in gas phase were mainly from primary emission during the biomass burning evenings and secondary formation during the clear days in rural China (Salvador et al., 2021). Our measurement site was several km away from biomass burning sites with ~7-10 km. And the 31 potential BrC in the gas-phase sum up to $8.5 \pm 6.7$ ng m$^{-3}$, significantly lower than 1720 ng m$^{-3}$ of 16 BrC (Salvador et al., 2021). Cheng et al. (2021) found that secondary formation was a strong source for five BrC molecules in the gas-phase. Therefore, BrC in the gas-phase are less influenced from primary emissions from biomass burning but are mainly controlled by secondary formation and partitioning in rural Germany.

**3.5 Diurnal variations and sources of potential BrC in the particle phase**

The 178 potential BrC molecules in the particle phase (PBrC) exhibited two peaks in the diurnal profile (Figure 4a) averaged over the whole winter campaign. They increased from 19:00 to 01:00 with a peak at $82 \pm 35$ ng m$^{-3}$ around midnight. Then the PBrC slowly decreased after midnight. However, they increased again from 6:00 to 08:00 and forming a second peak with $102 \pm 49$ ng m$^{-3}$ in the morning. During daytime, they decreased reaching lowest values with $61 \pm 31$ ng m$^{-3}$ at 14:00-15:00. During the nighttime and morning hours, the higher mass concentrations of PBrC were caused by residential wood burning emissions. Consistently, higher PM$_{2.5}$ concentration levels at nighttime at a rural site near Karlsruhe, Germany, could be assigned to wood burning emissions from wood stove operation during winter (Thieringer et al., 2022). The low mass concentrations of PBrC at daytime could be explained by photobleaching and evaporation of BrC, and/or dilution by the increasing planetary boundary layer heights (Satish et al., 2017). Satish et al. (2017) found that BrC over the Indo-Gangetic Plain had two peaks of BrC at evening and morning hours, and lowest values during daytime.

To determine the sources of brown carbon, we used the edge approach (Day et al., 2015). It allows to estimate the contribution of primary biomass burning (BB) to the measured BrC concentrations using levoglucosan as a primary source tracer. This approach is analogous to the widely used elemental carbon (EC) tracer approach, in which EC is used to distinguish the primary organic carbon (POC) and secondary organic carbon (SOC) in total organic carbon (OC) measurements (Day et al., 2015; Cabada et al., 2004). Levoglucosan (lev) and potential BrC were measured online by the same instruments and under the same conditions. As discussed above, we observed a good correlation (r = 0.8) between levoglucosan and BC during the winter campaign. Therefore, levoglucosan is a suitable tracer for primary BB. Please note that we did not calibrate the sensitivities of levoglucosan detected by FIGAERO-CIMS. Therefore, it could cause some uncertainties to estimate brown carbon from biomass burning and secondary formation. Figure 5a shows that the blue points can be used as edge points to determine the ratio of BrC/levoglucosan at the primary emissions from biomass burning. The relative contributions of primary emissions (BB) and secondary (sec) formation for total BrC molecules were estimated using the following expression:

$$BrC_{BB} = \left( \frac{[BrC]}{[lev]}_{BB} \right) * [lev.]$$

$$[BrC_{sec}] = [BrC_{Tot}] - [BrC_{BB}]$$

Where ([BrC]/[lev])$_{BB}$ is the ratio of the concentration of the BrC to that of levoglucosan in the primary emissions from
biomass burning and this value is 1.1 ± 0.1 (Figure 5a), BrC$_{BB}$ and BrC$_{sec}$ are the fractions of BrC generated through
biomass burning and secondary production, respectively, BrC$_{Tot}$ and lev. are the measured concentrations of BrC and
levoglucosan during the winter campaign. Using this approach, we calculated the diurnal profiles of BrC from primary
emissions (BrC$_{BB}$) and secondary formation (BrC$_{sec}$) shown in Figure 5b. The uncertainty of the splitting between BrC
from biomass burning and of secondary origin is mainly based on the levoglucosan concentration for which we have
included the calibration. Based on this we estimated the uncertainty of the BrC source splitting to ±35%. The mass
fraction of BrC$_{sec}$ increased at daytime and decreased at evening. This indicates that the secondary formation for BrC
in the particle phase was enhanced during daytime, facilitated by the higher levels of oxidants e.g. O$_3$ (Figure 4c). The
mass fraction of BrC$_{BB}$ had two peaks at early morning and in the evening hours, respectively. This may be caused by
residential wood burning emissions. BrC$_{BB}$ accounts for 39 ± 21% of the total BrC as averaged for the whole
measurement period. During biomass burning events, the BrC$_{BB}$ is a major mass fraction for total BrC that accounts
for 61 ± 13% during BB-event1 and 65 ± 12% during BB-event-2, respectively. Therefore, the primary emissions of
BrC have a significant impact on BrC, especially, at biomass burning events. However, on average over the whole
campaign, BrCsec dominates the mass fraction of BrC with 61 ± 21%. Therefore, the secondary formation can be
considered as an important source for BrC in rural Germany. Consistently, secondary formation from biomass burning
emission is important for the brown carbon absorption in the Switzerland, the central Europe. (Moschos et al., 2018).
Secondary sources for BrC were more important for absorption than primary ones in the Southeastern Margin of the
Tibetan Plateau (Wang et al., 2019b).
To further investigate the oxidation of BrC in the particle phase we plotted, the diurnal profiles of O/C ratios of BrC
during the whole campaign was measured, as shown in Figure 6. The O/C ratio of the potential BrC molecules increased
during daytime and decreased at nighttime. This is an indication for an impact of photo-oxidation on BrC either during
formation or aging leading to an increase of its O/C ratio. Consequently, the O/C ratio of the potential BrC molecules
shows a positive correlation (r = 0.8) with ozone, another product of photo chemistry. In contrast, the light absorption
of BrC at 370 nm (b$_{brc370}$) and the double bond equivalent (DBE) decreased at daytime and increased at nighttime.
During daytime, the absorption of brown carbon at 370 nm decreased due to lower DBE and higher O/C values of
brown carbon caused by photooxidation. This is in accordance with previous studies where atmospheric photooxidation
diminishes light absorption of primary brown carbon aerosol from biomass burning (Sumlin et al., 2017). Oxidative
whitening can reduce light absorption of brown carbon during the day (Hems et al., 2021).
**Conclusions**
The chemical composition, diurnal variation, and sources of brown carbon aerosol were investigated during February-
March 2021 in a rural area, at KIT Campus Nord, a location characteristic for central Europe. The 178 potential brown
carbon molecules (including 7 nitro aromatic compounds, NACs) identified in the particle phase contributed on average
$83 \pm 44$ ng m$^{-3}$ and 31 potential brown carbon molecules (including 4 NACs) identified in the gas phase contributed
on average $8.5 \pm 6.7$ng m$^{-3}$ during the whole campaign. During dedicated biomass burning events, potential BrC
concentrations in the particle phase were significantly higher with up to $\sim$254 ng m$^{-3}$. The 178 identified potential
brown carbon molecules only accounted for $2.6 \pm 1.5\%$ of the total organic mass, but explained $14 \pm 13\%$ of the total
brown carbon absorption at 370 nm, assuming a MAC$_{370}$ as 9.5 m$^2$ g$^{-1}$. This shows that a small fraction of the brown
carbon molecules dominates the overall absorption. This indicates the great importance of identifying these molecules,
the strong absorbers, to predict aerosol absorption.
Diurnal variations show that the particle-phase potential BrC had two peaks at early morning and evening hours,
respectively. These were mainly caused by residential wood burning emissions. In contrast, the gas-phase potential
BrC showed higher concentrations at daytime and lower concentrations at nighttime. The gas-phase BrC molecules
were mainly controlled by secondary formation (e.g. by photochemical processes) and particle-to-gas partitioning. The
two main sources contributed to particle-phase BrC were primary emission from biomass burning and secondary
formation. Secondary formation, e.g. by photooxidation, is an important source of particle-phase BrC corresponding
to increasing O/C ratios of BrC during daytime and a positive correlation (r = 0.8) with ozone concentrations. In
addition, the DBE of the particle-phase decreased during daytime. This indicates that the absorption of brown carbon
at 370 nm decreased due to lower DBE and higher O/C ratio due to the photooxidation of brown carbon. Compared
with previous measurements in central Europe (Lukács et al., 2007; Zhang et al., 2020), our study found that secondary
formation, e.g., photochemical processes, was an important source for BrC in gas and particle phases. To improve air
quality in winter, we need to reduce biomass burning emissions (e.g., regulate wood stoves) but also reduce the
precursors to form secondary aerosol. Overall, this study provides good insight into the light absorption, sources, and
diurnal variation from real-time observations of brown carbon molecules in central Europe by using mass spectrometry
and aethalometer.
*Data availability*
Data are available upon request to the corresponding author.
**Competing interests**
At least one of the (co-)authors is a member of the editorial board of Atmospheric Chemistry and Physics
**Author contributions**
FJ and HS designed the measurement campaign. FJ, LG, JS, and HS performed the experimental work. FJ did
FIGAERO-CIMS and AE33 data analysis. HS and HZ processed the trace gas and meteorological data, respectively.
TL gave general comments for this paper. FJ wrote the paper with contributions from all co-authors.
**ACKNOWLEDGMENTS**
The authors gratefully thank the staff of IMK-AAF for providing substantial technical support during the field
campaigns under COVID conditions. Furthermore, Feng Jiang and Junwei Song are thankful for the support from the
China Scholarship Council (CSC).

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

Contributions of nitrated aromatic compounds to the light absorption of water-soluble and particulate brown carbon in
different atmospheric environments in Germany and China, Atmos. Chem. Phys. 17, 1653-1672,
https://doi.org/10.5194/acp-17-1653-2017, 2017.

Thieringer, J. R. D., Szabadi, J., Meyer, J., and Dittler, A.: Impact of Residential Real-World Wood Stove Operation on Air Quality concerning PM2.5 Immission, Processes, 10, 545, https://doi.org/10.3390/pr10030545, 2022.

Thompson, S. L., Yatavelli, R. L. N., Stark, H., Kimmel, J. R., Krechmer, J. E., Day, D. A., Hu, W., Isaacman-VanWertz, G., Yee, L., Goldstein, A. H., Khan, M. A. H., Holzinger, R., Kreisberg, N., Lopez-Hilfiker, F. D., Mohr, C., Thornton, J. A., Jayne, J. T., Canagaratna, M., Worsnop, D. R., and Jimenez, J. L.: Field intercomparison of the gas/particle partitioning of oxygenated organics during the Southern Oxidant and Aerosol Study (SOAS) in 2013, Aerosol Sci. Technol. 51, 30-56, https://doi.org/10.1080/02786826.2016.1254719, 2017.

Wang, Q., Ye, J., Wang, Y., Zhang, T., Ran, W., Wu, Y., Tian, J., Li, L., Zhou, Y., Hang Ho, S. S., Dang, B., Zhang, Q., Zhang, R., Chen, Y., Zhu, C., and Cao, J.: Wintertime Optical Properties of Primary and Secondary Brown Carbon at a Regional Site in the North China Plain, Environ. Sci. Technol. https://doi.org/10.1021/acs.est.9b03406, 2019a.

Wang, Q. Y., Han, Y. M., Ye, J. H., Liu, S. X., Pongpiachan, S., Zhang, N. N., Han, Y. M., Tian, J., Wu, C., Long, X., Zhang, Q., Zhang, W. Y., Zhao, Z. Z., and Cao, J. J.: High Contribution of Secondary Brown Carbon to Aerosol Light Absorption in the Southeastern Margin of Tibetan Plateau, Geophys. Res. Lett. 46, 4962-4970, https://doi.org/10.1029/2019gl082731, 2019b.

Wu, G., Wan, X., Gao, S., Fu, P., Yin, Y., Li, G., Zhang, G., Kang, S., Ram, K., and Cong, Z.: Humic-like substances (HULIS) in aerosols of central Tibetan Plateau (Nam Co, 4730 m asl): Abundance, light absorption properties and sources, Environ. Sci. Technol. 52, 7203–7211, https://doi.org/10.1021/acs.est.8b01251, 2018.

Xie, M., Chen, X., Hays, M. D., Lewandowski, M., Offenberg, J., Kleindienst, T. E., and Holder, A. L.: Light Absorption of Secondary Organic Aerosol: Composition and Contribution of Nitroaromatic Compounds, Environ. Sci. Technol. 51, 11607– 11616, https://doi.org/10.1021/acs.est.7b03263, 2017.

Xu, J. Z., Hettiyadura, A. P. S., Liu, Y. M., Zhang, X. H., Kang, S. C., and Laskin, A.: Regional Differences of Chemical Composition and Optical Properties of Aerosols in the Tibetan Plateau, J. Geophys. Res.-Atmos., 125, e2019JD031226, https://doi.org/10.1029/2019jd031226, 2020.

Yang, Z., Tsona, N. T., George, C., and Du, L.: Nitrogen-Containing Compounds Enhance Light Absorption of Aromatic-Derived Brown Carbon, Environ. Sci. Technol. https://doi.org/10.1021/acs.est.1c08794, 2022.

Ye, C., B. Yuan, Y. Lin, Z. Wang, W. Hu, T. Li, W. Chen, C. Wu, C. Wang, S. Huang, J. Qi, B. Wang, C. Wang, W. Song, X. Wang, E. Zheng, J. E. Krechmer, P. Ye, Z. Zhang, X. Wang, D. R. Worsnop, and M. Shao.: Chemical characterization of oxygenated organic compounds in the gas phase and particle phase using iodide CIMS with FIGAERO in urban air, Atmos. Chem. Phys., 21, 8455-78, https://doi.org/10.5194/acp-21-8455-2021, 2021.

Yus-Díez, J., V. Bernardoni, G. Močnik, A. Alastuey, D. Ciniglia, M. Ivančič, X. Querol, N. Perez, C. Reche, M. Rigler, R. Vecchi, S. Valentini, and M. Pandolfi.: Determination of the multiple-scattering correction factor and its cross-sensitivity to scattering and wavelength dependence for different AE33 Aethalometer filter tapes: a multi-instrumental approach, Atmos. Meas. Tech., 14: 6335-55, https://doi.org/10.5194/amt-14-6335-2021, 2021.

Zeng, L. H., Zhang, A. X., Wang, Y. H., Wagner, N. L., Katich, J. M., Schwarz, J. P., Schill, G. P., Brock, C., Froyd,
K. D., Murphy, D. M., Williamson, C. J., Kupc, A., Scheuer, E., Dibb, J., and Weber, R. J.: Global Measurements of
Brown Carbon and Estimated Direct Radiative Effects, Geophys. Res. Lett. 47, https://doi.org/10.1029/2020gl088747,
617 2020.

Zhang, Y., Albinet, A., Petit, J.-E., Jacob, V., Chevrier, F., Gille, G., Pontet, S., Chrétien, E., Dominik-Sègue, M.,
Levigoureux, G., Močnik, G., Gros, V., Jaffrezo, J.-L., and Favez, O.: Substantial brown carbon emissions from
wintertime residential wood burning over France, Sci. Total Environ. 743, 140752,
https://doi.org/10.1016/j.scitotenv.2020.140752, 2020.
Zotter, P., H. Herich, M. Gysel, I. El-Haddad, Y. Zhang, G. Močnik, C. Hüglin, U. Baltensperger, S. Szidat, and A. S.
H. Prévôt. 2017.: Evaluation of the absorption Ångström exponents for traffic and wood burning in the Aethalometer-
based source apportionment using radiocarbon measurements of ambient aerosol, Atmos. Chem. Phys., 17, 4229-49,
https://doi.org/10.5194/acp-17-4229-2017, 2017.







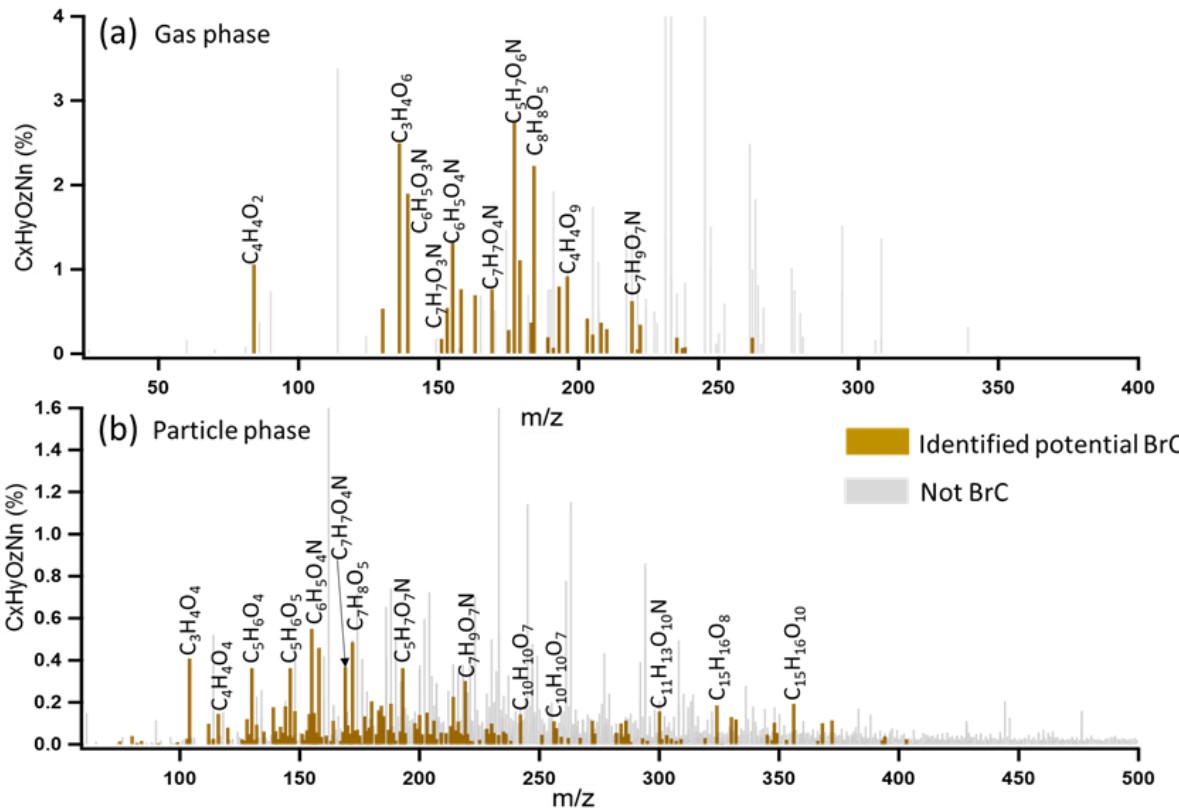


**Figure 1. CIMS mass spectra of organic aerosol measured by FIGAERO-CIMS for a biomass burning event on March 1st, 2021, a: gas phase, b: particle phase. The CI source employs reactions of I⁻ ions, which convert analyte molecules into [M+I]⁻ ions. Legends above MS features correspond to neutral molecules. The brown peaks in mass spectra were assigned as potential BrC molecules, while the gray peaks refer to the other organic molecules.**


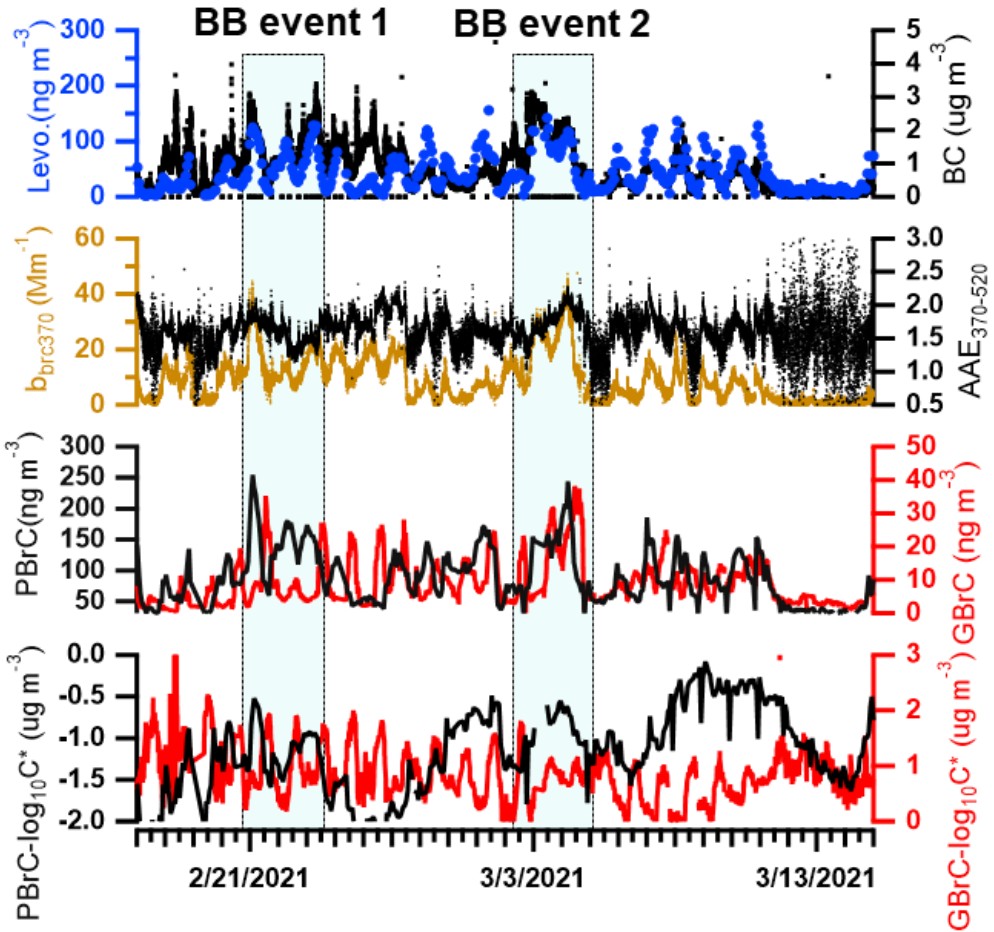


**Figure 2. Time series of levoglucosan (Levo.) concentrations in particle phase from FIGAERO-CIMS, BC concentrations from aethalometer (AE33), absorption of brown carbon at 370 nm (brc$_{370}$), absorption Ångström exponents between 370 nm and 520 nm (AAE$_{370-520}$), brown carbon concentrations in particle phase (PBrC) and gas phase (GBrC) and volatility (log$_{10}$C*) of brown carbon in particle phase (PBrC_log$_{10}$C*) and gas phase (GBrC_log$_{10}$C*) during the winter campaign.**



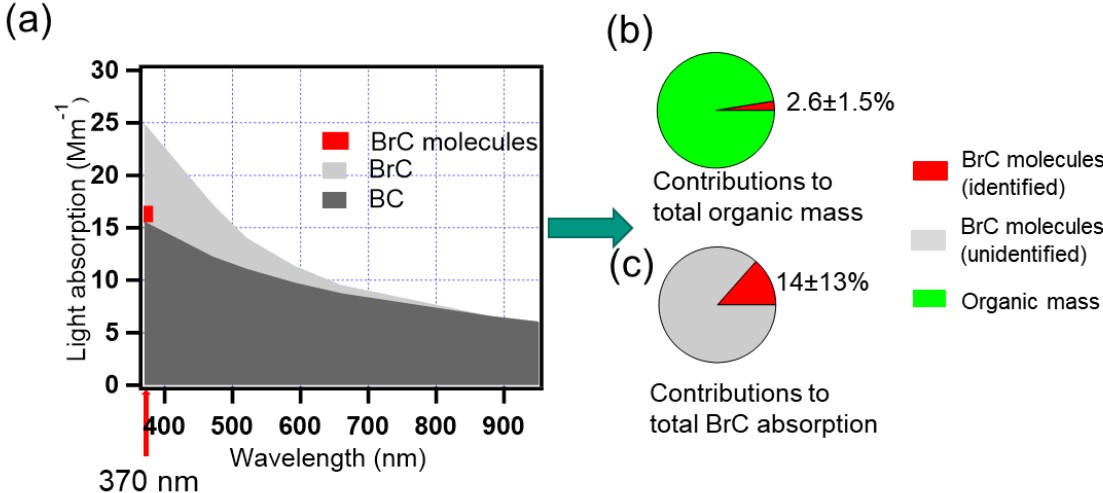


**Figure 3. (a) A stacked plot showing the main contributions to aerosol absorption from brown carbon and black**
**carbon based on the seven wavelengths measured by the aethalometer AE33. The contribution of the identified**
**brown carbon molecules to the total aerosol absorption is indicated in red at 370 nm. (b) Average mass**
**contribution of the potential BrC molecules to estimated total organic mass and (c) absorption contribution of**
**the potential BrC molecules identified to total absorption by BrC. The red pie: identified BrC molecules; the**
**gray pie: unidentifed-BrC molecules; the green pie: all organic mass.**


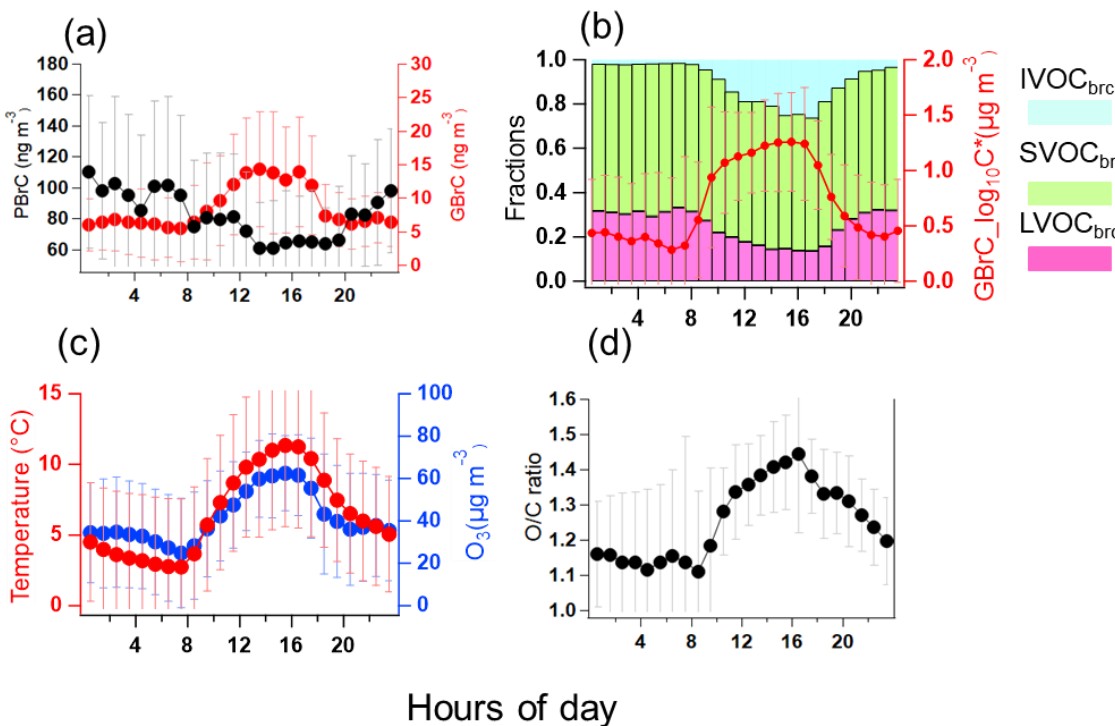


Figure 4. Diurnal profiles averaged over the whole winter campaign of (a) BrC in the particle (PBrC) and gas
phase (GBrC), (b) BrC volatility fractions in LVOC$_{brc}$, SVOC$_{brc}$, IVOC$_{brc}$, and mean BrC volatility in the gas
phase (red line), (c) temperature and ozone concentration. (d) O/C ratio of the oxidized organic components in
the gas phase.

662

663

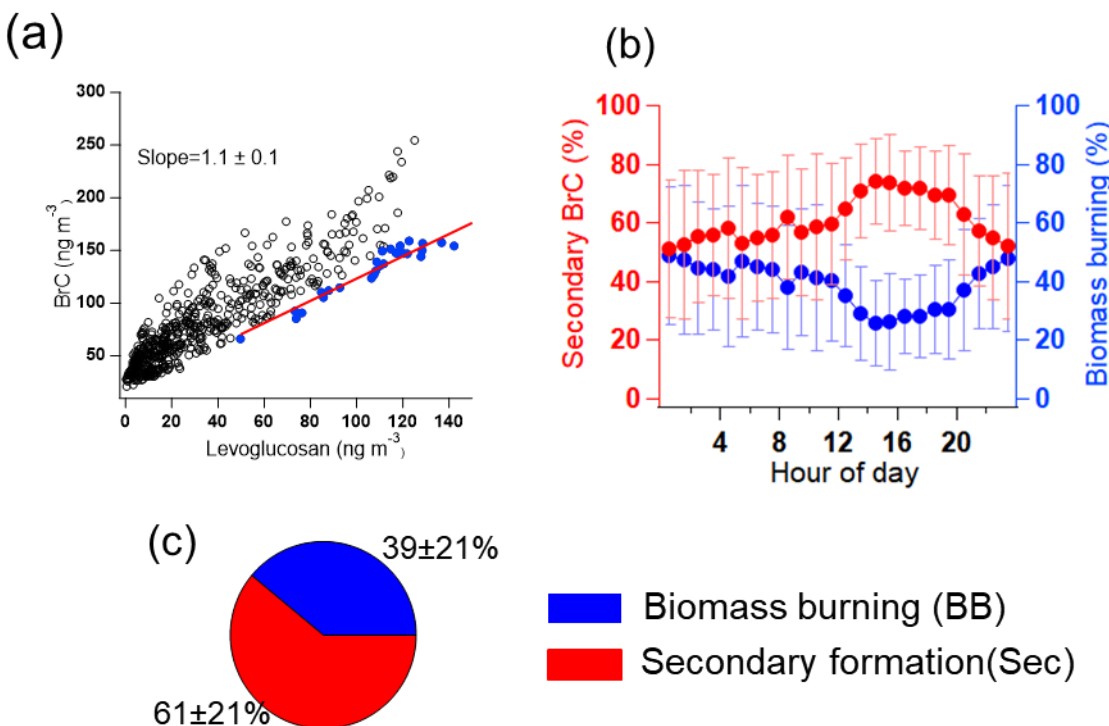

664

**Figure 5. (a) Correlation analysis of BrC and levoglucosan in the particle phase for the analysis of the**
**contribution of biomass burning using the edge method (Day et al., 2015). Blue points are the data used to**
**determine [BrC/lev.]ᴮᴮ. (b) diurnal profile of secondary-formation BrC and biomass-burning BrC for the whole**
**measurement campaign. (c) Average mass fractions of secondary formed BrC and biomass-burning primary**
**BrC for the whole campaign.**




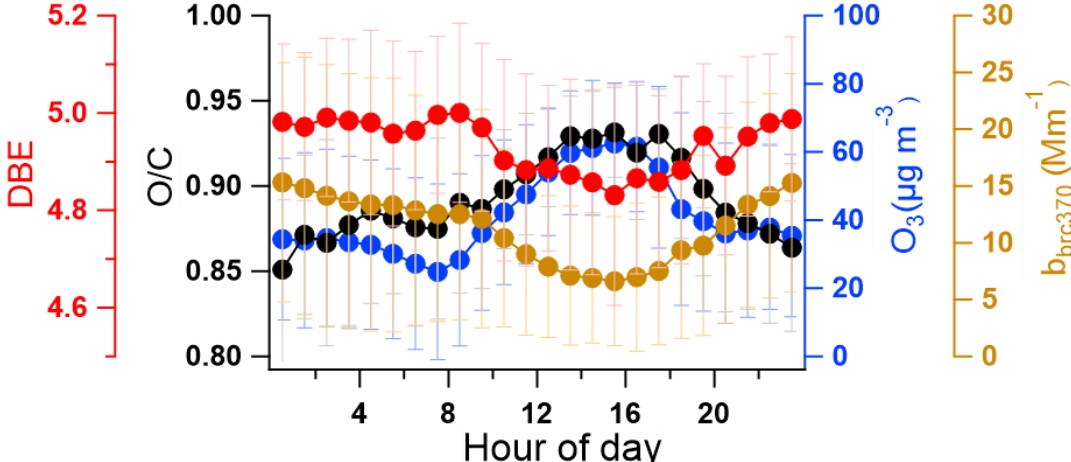


**Figure 6. The diurnal profile of DBE (double bond equivalent), O/C ratio of BrC, $O_3$, and $b_{brc370}$ (absorption of BrC at 370 nm) during the whole measured period.**




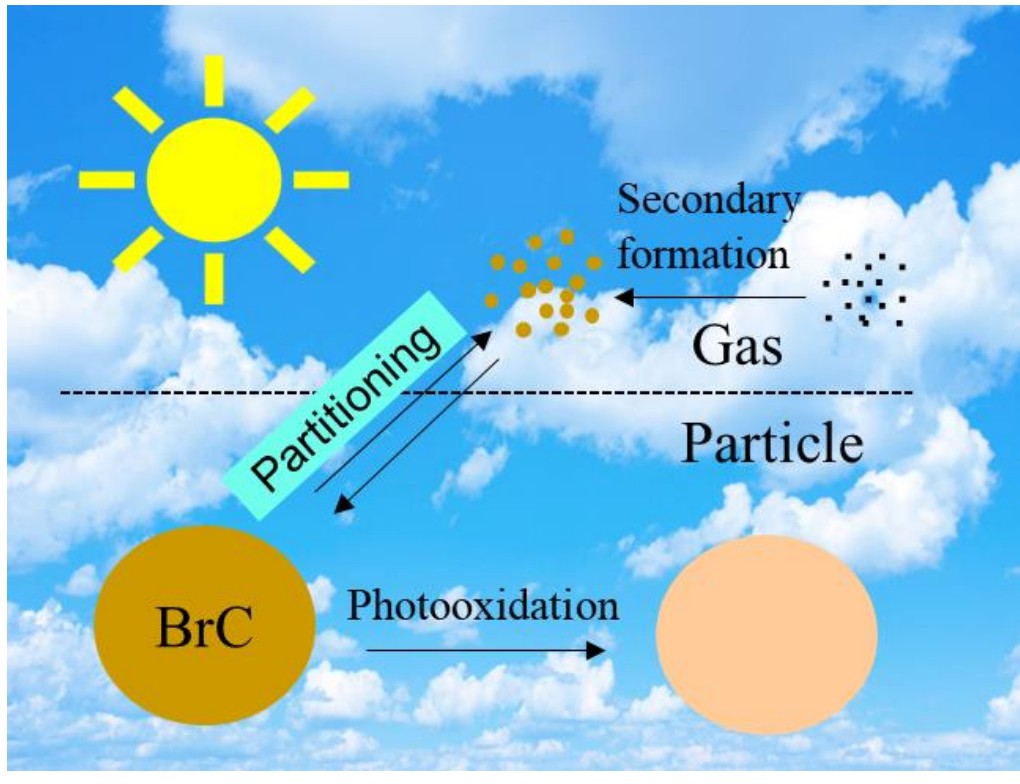


**Figure 7. A conceptual picture of the abstract**

