# Peer review of "Brown carbon aerosol in rural Germany: sources, chemistry, and"

_EGUsphere, 2024_

## Author Comment (AC1)

**Response to reviewers' comments on "Brown carbon aerosol in rural Germany: sources, chemistry, and diurnal variations" (egusphere-2024-1848)**

The authors kindly thank the reviews for the careful review of the manuscript, and the helpful comments and suggestions, which improve the manuscript a lot. All the comments are addressed below point by point, with our responses in blue, and the corresponding revisions to the manuscript in red. All updates of the original manuscript are marked in the revised version.

**Reviewer #1**

This paper describes ambient air measurements conduced for one month in winter in Germany in 2021. In addition to standard instrumentation for $O_3$, $NO_x$, particle sizes, etc, the key measurements were gas and particle phase composition using FIGAERO I-CIMS and an aethalometer. In addition to black carbon (BC) source apportionment, the goals of the work were to identify potential brown carbon (BrC) molecules in the gas and particle states, to study their diurnal variations, and to determine to what degree they contribute to the overall BrC measured by the aethalometer.

There is merit to this type of study because we need more molecular information about BrC molecules and their behavior, especially in regions which are not predominantly influenced by wildfire emissions, i.e., in this location the sources are presumably residential biomass burning and fossil fuel combustion. The findings are that there are both biomass burning (larger) and fossil fuel (smaller) contributions to BC, there were about 200 or so mass spectral features that may be BrC molecules, these features contribute about 10 percent to the total BrC absorption at 370 nm but a lower fraction of the total organic aerosol mass, gas phase BrC is largely photochemically generated during the day, and the ratio of gas phase BrC to particle phase BrC is much less than unity.

**General comments**

These are potentially interesting new measurements in Germany of quantities that have been measured at other locations. The methods and results are not particularly novel, and the paper needs to be much more quantitatively rigorous, especially with regard to uncertainties. I wonder whether this paper should be classified as an ACP "Measurement Report"? Overall, there is quite a bit of work to be done to get this paper ready for publication.

We modified the manuscript and added calibrations to make the results quantitative. In principle, we are open to convert our manuscript to a measurement report. However, we are not sure if this should be done at this stage and would like to know if the editor would advise us to do.

**Specific comments:**

1. My major criticism of this paper is the quantitative uncertainties in the measurements. In particular, unless I missed mention of them, there are no calibrations for the FIGAERO I-CIMS measurements. Rather, I believe that an "average sensitivity" was used for all mass spectral features, with the value taken from a literature study, i.e., with an entirely different instrument/operator. This is quite problematic, as CIMS instruments vary widely in sensitivity from one to another, even if operated in nominally the same manner. Calibration of at least a small

number of standard compounds is the minimum standard for field work, and increasingly many molecules are calibrated (or the voltage scanning method is applied) for I-CIMS work.

Indeed, the sensitivities of CIMS instruments can vary substantially and the sensitivity of the Iodide CIMS is known to vary over several orders of magnitude for different compounds. Application of an average, e.g. maximum, sensitivity can be useful to compare compounds for which standards are hardly available, like for highly oxygenated compounds. However, we agree that it is meaningful to calibrate our CIMS with some known BrC standard compounds. Therefore, we calibrated our instrument with four different nitro aromatic compounds (4-nitrophenol, 4-nitrocatechol, 4-methyl-5-nitrocatechol, and 2-methyl-4-nitrophenol).

"After the filed campaign, the calibration of 4-nitrophenol, 4-nitrocatechol, 2-methyl-4-nitropehnol, and 4-methyl-5-nitrocatechol was utilized to characterize the sensitivity factor of nitro aromatic compounds (NACs). Each NACs was dissolved into methanol to about 10 ng μL$^{-1}$ as a standard NACs solution. Different volume (1, 2, 5, and 10 µL) of the standard NACs solution was deposited on a PTFE filter using an accurate syringe. The deposited filter was heated by FIGAERO-iodide-CIMS with ultra-high purity nitrogen following a thermal desorption. The filters were then desorbed in the same way as for the field samples. Every volume of the standard solution was repeated three times. The sensitivity factors of our iodide CIMS for 4-nitrophenol, 4-nitrocatechol, 2-methyl-4-nitropehnol, and 4-methyl-5-nitrocatechol were $0.80 \pm 0.44$, $0.50 \pm 0.32$, $0.96 \pm 0.52$, $0.97 \pm 0.63$, respectively. The sensitivity factors are the ratios of measured concentrations by FIGAERO-CIMS from the calibration vs "standard concentrations", where measured concentrations were calculated using a nominal maximum sensitivity of 22 cps/ppt (Salvador et al., 2021). The average sensitivity factor of all four NACs was $0.81 \pm 0.53$. We used this average sensitivity factor to estimate the concentrations of other potential brown carbon molecules in this study."

[Figure]

Figure S9. Calibration of FIGAERO-CIMS with four different nitro aromatic compounds. Blue: 4-nitropehnol; Green: 4-nitrocatechol; Red: 2-methyl-4-nitropehnol; Black: 4-methyl-5-nitrocatechol.

2. Moreover, calibrations for particle bound species (such as levoglucosan) can be performed with the FIGAERO by depositing known amounts of these molecules on the collecting filter. Thus, the authors have to better justify their reports of absolute amounts of BrC molecules. If they have not calibrated themselves, I do not believe they can report an absolute amount.

We agree and calibrated our FIGAERO-CIMS with the important biomass burning tracer levoglucosan. The calibration method was same as NACs calibration. The sensitivity factor was determined to $0.40 \pm 0.14$ corresponding to sensitivity of $9 \pm 3$ cps/ppt.

We added a few sentences as follow:

"The sensitivity factor of levoglucosan was $0.40 \pm 0.14$ in this study (Figure S10). We used the sensitivity factor of $0.40 \pm 0.14$ to estimate the concentrations of molecules, which are not identified as potential BrC molecules."

[Figure]

Figure S10. A calibration of FIGAERO-CIMS with levoglucosan. The measurement concentrations are calculated using a maximum sensitivity of 22 cps/ppt (Salvador et al., 2021).

3. On a related note, the authors appear to dismiss this uncertainty after acknowledging it: Line 131 "These values have high uncertainty with several orders of magnitude. However, this is still a reasonable method to measure the organic aerosol in atmosphere." They need to justify why this approach is "reasonable".

We agree that this text is unclear and modified it as follows:

"In this study, BrC molecules were identified and partially quantified in atmospheric aerosol by FIGAERO-CIMS. Please note that the iodide CIMS has sensitivities varying over several orders magnitude for different compounds e.g. of different oxidation states (Lopez-Hilfiker et al., 2016).

Therefore, the quantitative interpretation is limited to the small amount of compounds for which we could do calibration with authentic standards. Keeping this in mind, it can still be meaning to a relative comparison of the large number of high oxidized compounds assuming the same sensitivity."

4. In an analogous manner, aethalometer measurements require care to interpret, with corrections for on-filter scattering and loadings. Although the authors mention these uncertainties, they do not provide quantitative estimates for them.

To address this point, we added the following sentences on the uncertainty and calibration of aethalometer measurements in section 2.4.

"Since our aethalometer has been used two loading spots, the loading effect was corrected by a Dual-spot loading compensation algorithm (Drinovec et al., 2015). To further address the scattering effect (Yus-Díez et al., 2021), we did comparison experiments in the Aerosol Preparation and Characterization (APC) chamber (Huang et al., 2018). Black carbon was injected into the APC chamber by using the PALAS soot generator (GfG 1000, Palas) (Saathoff et al., 2003). The APC chamber was connected to a photoacoustic spectrometer (PAS) operating at three wavelengths (405, 520, and 658 nm) (Linke et al., 2016) and an aethalometer AE33. As shown in Figure S11, for three wavelengths (370, 520, and 660 nm), the correlation slopes were 1.88, 1.94, and 1.98, respectively. The average multiple-scattering correction factor was $1.90 \pm 0.06$ in this study."

[Figure]

Fig. S11. Comparison of aerosol absorption coefficient measured by the photoacoustic spectrometer (PAS) and the aethalometer AE33 at 370 nm (a), (b), 520 nm, and 660 nm (c).

5. Likewise, the paper performs BC source apportionment, and it decouples BC absorption from total absorption to arrive at BrC absorption. There are many ways to do these analyses. The paper should justify the methods chosen.

We used the Aethalometer model to obtain BC associated with fossil fuel ($BC_{ff}$) and wood burning ($BC_{wb}$). The calculation function is given as below:

$$BC_{wb} = [\frac{b_{abs}(470nm) - b_{abs}(950nm) * \left(\frac{470}{950}\right)^{-a_{ff}}}{\left(\frac{470}{950}\right)^{-a_{wb}} - \left(\frac{470}{950}\right)^{-a_{ff}}}]/b_{abs}(950nm) * BC$$

The absorption Ångström exponents (α) for fossil fuel and wood burning were αff and αwb, respectively. One of the largest sources of uncertainty in the Aethalometer model is related to the section of αff and αwb values (Healy et al., 2017; Zotter et al., 2017). In addition, the αff was typically in the range of ∼ 0.8 – 1.2 in ambient air whereas αwb can vary from 1.6 to 2.2 (Saarikoski et al., 2021). However, we used the αff and αwb values as 0.95 and 1.68 to calculate the BC source (Helin et al., 2018), since our measurement site is in a rural area and nearby a suburban area.

We included this as follow:

"One of the largest sources of uncertainty in the Aethalometer model is related to the section of αff and αwb values (Healy et al., 2017; Zotter et al., 2017). In addition, the αff was typically in the range of ∼ 0.8 – 1.2 in ambient air whereas αwb can vary from 1.6 to 2.2 (Saarikoski et al., 2021). However, we used the αff and αwb values as 0.95 and 1.68 to calculate the BC source (Helin et al., 2018), since our measurement site is in a rural area and nearby a suburban area."

We added some sentences to describe how to determine BrC absorption:
"We assumed that the absorption from dust and other aerosol was negligible. Hence, the absorption was only contributed from BC and BrC. Therefore, Abs(λ) can be divided in BC and BrC absorption:

$$Abs = Abs_{BrC}(\lambda) + Abs_{BC}(\lambda) \tag{1}$$

where $Abs_{BrC}(\lambda)$ is the absorption caused by BrC at the following aethalometer wavelengths, $\lambda$ = 370, 470, 520, 590, or 660 nm while $Abs_{BC}(\lambda)$ is the absorption contributed by BC at the same wavelength (Wang et al., 2019). To determine $Abs_{BC}(\lambda)$ at each wavelength, we assumed that BC was the only absorber at $\lambda$ = 880 nm, and thus the $Abs_{BC}(\lambda)$ ($\lambda$ = 370, 470, 520, 590, and 660) can be extrapolated from the following equation:

$$Abs_{BC}(\lambda) = Abs_{880} \times \left(\frac{\lambda}{880}\right)^{-AAE_{BC}} \tag{2}$$

where $AAE_{BC}$ represents the spectral dependence of $Abs_{BC}(\lambda)$, and a value of 1.0 was chosen for $AAE_{BC}$ based on previous studies in Germany (Teich et al., 2017). Finally, one can obtain the $Abs_{BrC}(\lambda)$ as follows:

$$Abs_{BrC}(\lambda) = Abs(\lambda) - Abs(880) \times \left(\frac{\lambda}{880}\right)^{-AAE_{BC}} \tag{3}$$

6. Moreover, it uses a literature value for the MAC value of BrC. How variable are these values from one site to another? The MAC value could be strongly dependent on the type of BrC being analyzed.
The mass absorption coefficients of potential BrC molecules vary substantially from molecule to molecules (Moschos et al., 2024). Indeed, some weakly absorbing compounds may sometimes contribute with a higher mass fraction and some highly absorbing compounds may dominate the absorption despite of small mass fractions at other locations. In order to estimate the potential light absorption of the 171 potential BrC molecules, we assumed an average MAC value for them in a similar way as we have done previously (Jiang et al., 2022). This allows to estimate the order of magnitude of the potential BrC absorption.

To make this clear we have modified the text in section 3.3 as follows:

"In order to calculate the light absorption from the other 171 potential brown carbon molecules identified, we assumed an average MAC value of 9.5 m$^2$g$^{-1}$ at 370 nm for all BrC molecules to estimate their absorption (Jiang et al., 2022). So far, the MAC$_{370}$ of most potential brown carbon molecules are still unknown. In addition, since the potential BrC molecules detected by FIGAERO-CIMS could have isomers, we did not calibrate mass absorption coefficients of 171 potential BrC. Despite these uncertainties, we think it is reasonable to estimate the order magnitude of the total BrC absorption based on this assumption."

7. What uncertainties are there in the total organic aerosol mass loading given that there was no measurement of it during the campaign?

Unfortunately, our aerosol mass spectrometer (HR-ToF-AMS, Aerodyne) was not available due to a technical problem during the time of this measurement campaign. Therefore, we have no independent quantification of the total organic aerosol mass loadings. However, we estimated the total organic mass as a fraction of 50 ± 20 % of PM$_{2.5}$ which is a typical fraction for this season and region at the location in Germany (Song et al., 2022; Song et al., 2024). According to this assumption, the organic mass detected by FIGAERO-CIMS based on calibrated sensitivity factors was 37 ± 20 % of the estimated total organic mass. This is in a similar range as observed in previous studies (Ye et al., 2021). We modified the manuscript text as follow:

"We have no independent quantification of the total organic aerosol mass loadings. However, we estimated the total organic mass as a fraction of 50 ± 20 % of PM$_{2.5}$ which is a typical fraction for at the location (Song et al., 2022; Song et al., 2024). According to this assumption, the average organic mass concentration was 4.5 ± 3.1 µg m$^{-3}$. The organic mass detected by FIGAERO-CIMS based on calibrated sensitivity factors was 37 ± 20% of the estimated organic mass. This is in a similar range as observed in previous studies (Gao et al., 2022; Ye et al., 2021)."

8. The paper does not provide a justification for how BrC molecules are identified from 1000's of mass spectral features, aside from providing a reference. How accurate are the mass fittings and the calculation of DBE and elemental composition for each feature? In other words, are these fittings unique for only one elemental formula? The paper should identify the BrC mass spectral features identified, with some indications of their intensities. Were any mass spectral features observed in both the gas and particle phase spectra? It would be interesting to know this, and a partition coefficient could be calculated.

Lin et al., (2018) assigned potential brown carbon compounds in the plot of DBE vs the number of carbon and nitrogen atoms per molecule (Figure S12). They employed high-resolution mass spectrometry to analyze biomass burning organic aerosol. We used this method to assign 178 potential BrC molecules (including 7 NACs) in the particle phase and 31 potential BrC molecules (including 4 NACs) in the gas phase. Figure 1 shows the mass spectra of organic aerosol detected by FIGAERO-CIMS in the gas and particle phase. The partitioning coefficients of potential BrC molecules are given in Table S6.

We added the following sentences to the text:

"Lin et al., (2016, 2018) employed high-resolution mass spectrometry to analyze biomass burning organic aerosol. They assigned potential brown carbon compounds according to the correlation of double bond equivalents (DBE) with the number of carbon atoms per molecule (Figure S12). We used this method to assign 178 potential BrC molecules (including 7 NACs) in the particle phase and 31 potential BrC molecules (including 4 NACs) in the gas phase, as shown in Figure 1 in the corresponding mass spectra. The gas to particle phase partitioning coefficients of those semi volatile potential brown carbon molecules which could be measured in both phases with sufficient sensitivity are listed in table S6."

[Figure]

Figure S12. Plot of the double bond equivalent (DBE) vs numbers of carbon and nitrogen atoms according to our measurements following the procedure described by Lin et al., (2018). The lines indicate DBE reference values of linear conjugated polyenes (red solid line) and fullerene-like hydrocarbons with DBE=0.9*C (black solid line). Data points inside the yellow shaded area are potential BrC molecules. (cf. Lin et al., 2018).

[Figure]

Figure 1. CIMS mass spectra of organic aerosol measured by FIGAERO-CIMS for a biomass burning event on March 1$^{st}$, 2021, a: gas phase, b: particle phase. The CI source employs reactions of I$^-$ ions, which convert analyte molecules into [M+I]$^-$ ions. Legends above MS features correspond to neutral molecules. The brown peaks in mass spectra were assigned as potential BrC molecules, while the gray peaks refer to the other organic molecules.

Table S6. Gas to particle phase partitioning coefficients of those semi volatile potential brown carbon molecules which could be measured in both phases with sufficient sensitivity (average and standard deviation).

| Number | Formula | Average g/p ratio | Std g/p ratio |
|--------|---------|-------------------|---------------|
| 1 | C3H5O6N1 | 0.81 | 0.67 |
| 2 | C4H4O2 | 9.7 | 10.1 |
| 3 | C4H5O6N1 | 2.1 | 1.8 |
| 4 | C4H5O7N1 | 1.8 | 1.8 |
| 5 | C5H5O8N1 | 0.8 | 0.5 |
| 6 | C5H6O4 | 0.2 | 0.4 |
| 7 | C5H7O6N1 | 3.2 | 4.4 |
| 8 | C5H7O7N1 | 0.3 | 0.3 |
| 9 | C6H6O5 | 0.1 | 0.1 |
| 10 | C6H6O10 | 1.3 | 0.9 |
| 11 | C6H7O6N1 | 0.4 | 0.3 |
| 12 | C6H7O7N1 | 0.6 | 0.6 |
| 13 | C6H7O8N1 | 0.8 | 0.4 |
| 14 | C6H5O3N1 | 0.5 | 2.1 |
| 15 | C6H5O4N1 | 0.1 | 0.1 |
| 16 | C7H7O3N1 | 0.7 | 0.4 |
| 17 | C7H7O4N1 | 0.3 | 0.2 |
| 18 | C7H9O6N1 | 0.4 | 0.3 |
| 19 | C7H9O7N1 | 0.5 | 0.6 |
| 20 | C7H9O8N1 | 0.4 | 0.4 |
| 21 | C8H8O5 | 0.5 | 0.3 |
| 22 | C8H9O4N1 | 0.3 | 0.2 |
| 23 | C9H10O9 | 0.6 | 0.4 |

9. Can a non-parametric wind direction analysis be provided to aid source apportionment?

Such an analysis can of course be an additional indicator for certain sources. However, for our wind measurements, we see a strong impact of channeling by the buildings nearby which inhibits a useful analysis.

[Figure]

Figure. Non-parametric wind regression (NWR) plots of particle-phase BrC (a), BC (b), and gas-phase BrC (c), respectively. Note: different scales of concentration are used, BrC: ng m$^{-3}$, BC: µg m$^{-3}$.

10. Line 267.   Photochemical activity forms ozone and is not the only cause of aging.

We agree and reformulated the text as follows:

"Therefore, the higher fraction of IVOC in the gas phase at daytime is most likely caused by secondary formation e.g. photochemical conversion/aging because of higher oxidant levels as indicated e.g. by higher concentration of ozone at same time (Figure 4c) (Saarikoski et al., 2021)."

References:

Drinovec, L., G. Mocnik, P. Zotter, A. S. H. Prevot, C. Ruckstuhl, E. Coz, M. Rupakheti, J. Sciare, T. Muller, A. Wiedensohler, and A. D. A. Hansen.: The "dual-spot" Aethalometer: an improved measurement of aerosol black carbon with real-time loading compensation, Atmos. Meas. Tech., 8, 1965-79, https://doi.org/10.5194/amt-8-1965-2015, 2015.

Healy, R. M., U. Sofowote, Y. Su, J. Debosz, M. Noble, C. H. Jeong, J. M. Wang, N. Hilker, G. J. Evans, G. Doerksen, K. Jones, and A. Munoz.: Ambient measurements and source apportionment of fossil fuel and biomass burning black carbon in Ontario, Atmos. Environ., 161, 34-47, https://doi.org/10.1016/j.atmosenv.2017.04.034, 2017.

Helin, Aku, Jarkko V. Niemi, Aki Virkkula, Liisa Pirjola, Kimmo Teinilä, John Backman, Minna Aurela, Sanna Saarikoski, Topi Rönkkö, Eija Asmi, and Hilkka Timonen.: Characteristics and source apportionment of black carbon in the Helsinki metropolitan area, Finland, Atmos. Environ., 190, 87-98, https://doi.org/10.1016/j.atmosenv.2018.07.022, 2018.

Huang, Wei, Harald Saathoff, Aki Pajunoja, Xiaoli Shen, Karl-Heinz Naumann, Robert Wagner, Annele Virtanen, Thomas Leisner, and Claudia Mohr.: alpha-Pinene secondary organic aerosol at low temperature: chemical composition and implications for particle viscosity, Atmos. Chem. Phys., 18, 2883-98, https://doi.org/10.5194/acp-18-2883-2018, 2018.

Jiang, F., J. W. Song, J. Bauer, L. Y. Gao, M. Vallon, R. Gebhardt, T. Leisner, S. Norra, and H. Saathoff.: Chromophores and chemical composition of brown carbon characterized at anurban kerbside by excitation-emission spectroscopy and mass spectrometry, Atmos. Chem. Phys., 22, 14971-86, https://doi.org/10.5194/acp-22-14971-2022, 2022.

Lin, P., P. K. Aiona, Y. Li, M. Shiraiwa, J. Laskin, S. A. Nizkorodov, and A. Laskin.: Molecular Characterization of Brown Carbon in Biomass Burning Aerosol Particles, Environ. Sci. Technol., 50, 11815-24, https://doi.org/10.1021/acs.est.6b03024, 2016.

Lin, P., L. T. Fleming, S. A. Nizkorodov, J. Laskin, and A. Laskin.: Comprehensive Molecular Characterization of Atmospheric Brown Carbon by High Resolution Mass Spectrometry with Electrospray and Atmospheric Pressure Photoionization, Anal. Chem., 90, 12493-502, https://doi.org/10.1021/acs.analchem.8b02177, 2018.

Linke, C., I. Ibrahim, N. Schleicher, R. Hitzenberger, M. O. Andreae, T. Leisner, and M. Schnaiter.: A novel single-cavity three-wavelength photoacoustic spectrometer for atmospheric aerosol research, Atmos. Meas. Tech., 9, 5331-46, https://doi.org/10.5194/amt-9-5331-2016, 2016.

Lopez-Hilfiker, F. D., S. Iyer, C. Mohr, B. H. Lee, E. L. D'Ambro, T. Kurten, and J. A. Thornton. 2016.: Constraining the sensitivity of iodide adduct chemical ionization mass spectrometry to multifunctional organic molecules using the

collision limit and thermodynamic stability of iodide ion adducts, Atmos. Meas. Tech., 9, 1505-12, https://doi.org/10.5194/amt-9-1505-2016, 2016.

Moschos, Vaios, Cade Christensen, Megan Mouton, Marc N. Fiddler, Tommaso Isolabella, Federico Mazzei, Dario Massabò, Barbara J. Turpin, Solomon Bililign, and Jason D. Surratt.: Quantifying the Light-Absorption Properties and Molecular Composition of Brown Carbon Aerosol from Sub-Saharan African Biomass Combustion, Environ. Sci. Technol., 58, 4268-80. https://doi.org/10.1021/acs.est.3c09378, 2024.

Saarikoski, S., J. V. Niemi, M. Aurela, L. Pirjola, A. Kousa, T. Ronkko, and H. Timonen.: Sources of black carbon at residential and traffic environments obtained by two source apportionment methods, Atmos. Chem. Phys., 21, 14851-69, https://doi.org/10.5194/acp-21-14851-2021, 2021.

Saathoff, H., K. H. Naumann, M. Schnaiter, W. Schöck, O. Möhler, U. Schurath, E. Weingartner, M. Gysel, and U. Baltensperger.: Coating of soot and (NH4)2SO4 particles by ozonolysis products of α-pinene, J. Aerosol Sci., 34, 1297-321, https://doi.org/10.1016/S0021-8502(03)00364-1, 2003.

Salvador, C. M. G., R. Z. Tang, M. Priestley, L. J. Li, E. Tsiligiannis, M. Le Breton, W. F. Zhu, L. M. Zeng, H. Wang, Y. Yu, M. Hu, S. Guo, and M. Hallquist.: Ambient nitro-aromatic compounds - biomass burning versus secondary formation in rural China, Atmos. Chem. Phys., 21, 1389-406, https://doi.org/10.5194/acp-21-1389-2021, 2021.

Song, J., H. Saathoff, F. Jiang, L. Gao, H. Zhang, and T. Leisner.: Sources of organic gases and aerosol particles and their roles in nighttime particle growth at a rural forested site in southwest Germany, Atmos. Chem. Phys., 24, 6699-717, https://doi.org/10.5194/acp-24-6699-2024, 2024.

Song, J. W., H. Saathoff, L. Y. Gao, R. Gebhardt, F. Jiang, M. Vallon, J. Bauer, S. Norra, and T. Leisner.: Variations of PM2.5 sources in the context of meteorology and seasonality at an urban street canyon in Southwest Germany, Atmos. Environ., 282, 119147, https://doi.org/10.1016/j.atmosenv.2022.119147, 2022.

Teich, M., D. van Pinxteren, M. Wang, S. Kecorius, Z. B. Wang, T. Muller, G. Mocnik, and H. Herrmann.: Contributions of nitrated aromatic compounds to the light absorption of water-soluble and particulate brown carbon in different atmospheric environments in Germany and China, Atmos. Chem. Phys., 17, 1653-72, https://doi.org/10.5194/acp-17-1653-2017, 2017.

Wang, Qiyuan, Jianhuai Ye, Yichen Wang, Ting Zhang, Weikang Ran, Yunfei Wu, Jie Tian, Li Li, Yaqing Zhou, Steven Sai Hang Ho, Bo Dang, Qian Zhang, Renjian Zhang, Yang Chen, Chongshu Zhu, and Junji Cao.: Wintertime Optical Properties of Primary and Secondary Brown Carbon at a Regional Site in the North China Plain, Environ. Sci. Technol., 21, 12389-12397, https://doi.org/10.1021/acs.est.9b03406, 2019.

Ye, C., B. Yuan, Y. Lin, Z. Wang, W. Hu, T. Li, W. Chen, C. Wu, C. Wang, S. Huang, J. Qi, B. Wang, C. Wang, W. Song, X. Wang, E. Zheng, J. E. Krechmer, P. Ye, Z. Zhang, X. Wang, D. R. Worsnop, and M. Shao.: Chemical

characterization of oxygenated organic compounds in the gas phase and particle phase using iodide CIMS with FIGAERO in urban air, Atmos. Chem. Phys., 21, 8455-78, https://doi.org/10.5194/acp-21-8455-2021, 2021.

Yus-Díez, J., V. Bernardoni, G. Močnik, A. Alastuey, D. Ciniglia, M. Ivančič, X. Querol, N. Perez, C. Reche, M. Rigler, R. Vecchi, S. Valentini, and M. Pandolfi.: Determination of the multiple-scattering correction factor and its cross-sensitivity to scattering and wavelength dependence for different AE33 Aethalometer filter tapes: a multi-instrumental approach, Atmos. Meas. Tech., 14: 6335-55, https://doi.org/10.5194/amt-14-6335-2021, 2021.

Zotter, P., H. Herich, M. Gysel, I. El-Haddad, Y. Zhang, G. Močnik, C. Hüglin, U. Baltensperger, S. Szidat, and A. S. H. Prévôt. 2017.: Evaluation of the absorption Ångström exponents for traffic and wood burning in the Aethalometer-based source apportionment using radiocarbon measurements of ambient aerosol, Atmos. Chem. Phys., 17, 4229-49, https://doi.org/10.5194/acp-17-4229-2017, 2017.

---

## Author Comment (AC2)

**Response to reviewers' comments on "Brown carbon aerosol in rural Germany: sources, chemistry, and diurnal variations" (egusphere-2024-1848)**

The authors kindly thank the reviews for the careful review of the manuscript, and the helpful comments and suggestions, which improve the manuscript a lot. All the comments are addressed below point by point, with our responses in blue, and the corresponding revisions to the manuscript in red. All updates of the original manuscript are marked in the revised version.

**Reviewer #2**

Review of Jiang et al. Brown carbon aerosol in rural Germany: sources, chemistry, and diurnal variations. This manuscript presents results from concurrent measurements of aerosol chemical composition and light absorption from black carbon and light-absorbing organic aerosols (aka brown carbon) at a rural site in Germany during one month in winter 2021. The absorption apportionment and chemical speciation of brown carbon aerosols in both gas and particle phases are reported. The sources of brown carbon in rural Germany were identified based on its diurnal variability and regression analysis of brown carbon with emission tracers. In general, this study adds to the literature on the characteristics of brown carbon aerosols in rural Germany, but for the reasons outlined below I cannot recommend publication of this manuscript in its current form.

**General comments**

1. Overall, I am not convinced that the 178 molecules identified by FIGAERO-CIMS are representative of the brown carbon aerosols, not only because they contributed to a very small fraction (2%) of total organic mass as well as brown carbon absorption (11%), but also the correlation between the molecule mass and brown carbon absorption is not so good. There are a lot of scatters in Figure S6 which means most of the brown carbon absorption cannot be explained by the identified molecules. The authors also did not provide details in how these compounds are identified as brown carbon molecules rather than referencing to an earlier publication from the same group. In several places throughout the manuscript the authors simply refer the 178 molecules as particle-BrC and the 31 molecules as gas-BrC (e.g. Figure 1, Figure S6 and related text), which is not accurate given the reasons provided above.

From the 1500 molecules we observed we assigned 178 as potential BrC molecules according to a method published by Lin et al., (2018). They correlated the number of double bond equivalents with the number of carbon atoms per molecule. A few studies used this method to assign brown carbon molecules. For example, there are good correlations (r = 0.9) between mass absorption efficiency at 365 nm and potential brown carbon molecules of larger molecular weight (Tang et al., 2020). Xu et al., (2020) used this method to assign 149 nitrogen-containing potential BrC chromophores in the Tibetan Plateau and we used this method to assign potential BrC molecules in downtown Karlsruhe (Jiang et al., 2022). It is not unusual that only a small mass fraction of absorbing molecules can dominate the aerosol absorption (Mohr et al., 2013). We consider the correlation of the BrC absorption at 370 nm with the estimated BrC mass of 0.7 ± 0.1 shown in figure S6 as relatively good considering the underlying assumptions.

To better point this out we changed "BrC molecules" to "potential BrC molecules" throughout the manuscript and added additional explanations on the method in sections 2.3 and 3.2 as follows:

Section 2.3:
"Lin et al., (2016, 2018) employed high-resolution mass spectrometry to analyze biomass burning organic aerosol. They assigned potential brown carbon compounds according to the correlation of double bond equivalents (DBE) with the number of carbon atoms per molecule (FigureS12). We used this method to assign 178 potential BrC molecules (including 7 NACs) in the particle phase and 31 potential BrC molecules (including 4 NACs) in the gas phase, as shown in Figure 1 in the corresponding mass spectra. A few other studies used this method also to assign more brown carbon molecules. For example, good correlations (r = 0.9) between mass absorption efficiency at 365 nm and potential brown carbon molecules of larger molecular weight were found by Tang et al., (2020). Xu et al., (2020) used this method to assign 149 nitrogen-containing potential BrC chromophores at the Tibetan Plateau and we used this method to assign potential BrC molecules in downtown Karlsruhe (Jiang et al., 2022). The potential BrC molecules we assigned according to this method for the particle and the gas phase are listed in Tables S2 and S3."

[Figure]

Figure S12. Plot of the double bond equivalent (DBE) vs numbers of carbon and nitrogen atoms according to our measurements following the procedure described by Lin et al., (2018). The lines indicate DBE reference values of linear conjugated polyenes (red solid line) and fullerene-like hydrocarbons with DBE=0.9*C (black solid line). Data points inside the yellow shaded area are potential BrC molecules. (cf. Lin et al., 2018).

[Figure]

Figure 1. CIMS mass spectra of organic aerosol measured by FIGAERO-CIMS for a biomass burning event on March 1st, 2021, a: gas phase, b: particle phase. The CI source employs reactions of $I^-$ ions, which convert analyte molecules into $[M+I]^-$ ions. Legends above MS features correspond to neutral molecules. The brown peaks in the mass spectra were assigned as potential BrC molecules while the gray peaks refer to the other organic molecules.

Section 3.2:
"We identified 178 potential BrC molecules according to the method developed by Lin et al., (2018) (cf. section 2.3.). The mass of these molecules shows a good correlation (r=0.7 ± 0.1) with the absorption at 370 nm ($b_{BrC370}$) of BrC (sf. Figure S6). "

2. There is lack of a discussion on the uncertainties related to absorption measurement by the aethalometer and calculations deriving BC and BrC absorption, as well as BrC source apportionment in Line 302-303. In Line 190-191, the authors stated that "During this winter campaign, the BrC absorption accounted for ~40% of total absorption caused by BC and BrC." I could not trace back to how this number (40%) was derived, nor did the authors provide information about which absorption wavelength the calculation is based on.

BC and BrC absorption measurements by aethalometers have the filter-based lensing effect (Moschos et al. 2021). According to previous studies, the lensing effect for BC and BrC measurement were 8%-27% and 6%-20%, respectively (Moschos et al. 2021). We adopted an $AAE_{BC}$ value of 1 in this study. However, this assumption introduces an uncertainty in the estimations of BC and BrC light absorptions. According to previous studies, the $AAE_{BC}$ shows a range of 0.8-1.4 (Lack and Langridge 2013). This range, although maybe not fully applicable to

our measurement location, potentially causes relatively large uncertainties in splitting between BrC and BC absorption (Duan et al. 2024).

We added following sentences to make the reader aware of this problem:
"The absorption measurements by aethalometer have the filter-based lensing effect (Moschos et al. 2021). According to previous studies, the uncertainty from lensing effect for BC and BrC measurement were 8%-27% and 6%-20%, respectively (Moschos et al. 2021). We assumed an $AAE_{BC}$ value of 1.0 in this study. However, this assumption introduces an uncertainty in the estimations of BC and BrC light absorptions. According to previous studies, the $AAE_{BC}$ ranges between 0.8-1.4 (Lack and Langridge 2013). This range although maybe not fully applicable to our measurement location, potentially causes relatively large uncertainties of up 81% (at 370nm) in splitting between BrC and BC absorption (Figure S13) (Duan et al. 2024). Despite these potentially large uncertainties on absolute absorption values, we consider this method still useful. Our assumption of $AAE_{BC} = 1.0$ is reasonable for our location as based on previous measurements and it should still allow to discuss the relative evolution of BC and BrC absorption."

[Figure]

Figure S13. Light absorption of BC (a) and BrC (b) under different assumptions regarding the $AAE_{BC}$. The blue, yellow, red, and black makers represent light absorption of BC and BrC when AAEBC is 1.4, 1.2, 1.0, and 0.8, respectively.

"The uncertainty of the splitting between BrC from biomass burning and of secondary origin is mainly based on the levoglucosan concentration for which we have included the calibration. Based on this we estimated the uncertainty of the BrC source splitting to ±35%."

We explained that we calculated the BrC absorption contribution for a wavelength at 370 nm and indicated this in the manuscript as follow:

"During this winter campaign, the BrC absorption accounted on average for ~40% of total absorption caused by BC and BrC at 370 nm."

3. The authors also tend to draw causal relationship based on correlation. For example, in Line 200, the authors stated "The levoglucosan had a good correlation (r=0.7) with BC. This also indicates that BC was mainly emitted from biomass burning during the winter campaign." This statement is not supported by evidence. Having a r =0.7 means levoglucosan can explain less than half of the variability in BC concentration. Similar statement is in Line 322-324 "In addition, the O/C ratio of BrC had a positive correlation (r=0.8) with ozone. This indicates that the BrC was photo-oxidized leading to an increase of the O/C ratio of BrC."

Thank for pointing to this. It wasn't our intention to draw causal relationship based on correlations.

We reformulated the sentence in line 200 as follows:

"The levoglucosan showed a good correlation (r = 0.7) with BC. This is in line with the large fraction of biomass burning contributing to BC during the winter campaign. Biomass burning BC accounted for (71 ± 40)% of total BC as we discussed above."

We reformulated the sentence in line 322-324 as follows:

"The O/C ratio of the potential BrC molecules increased during daytime and decreased at nighttime. This is an indication for an impact of photo-oxidation on BrC either during formation or aging leading to an increase of its O/C ratio. Consequently, the O/C ratio of the potential BrC molecules shows a positive correlation (r=0.8) with ozone, another product of photo chemistry."

**Specific comments:**

4. In Figure 2, the contribution from nitro-aromatics absorption is only plotted at 370 nm. I wonder if the absorption profile of these compounds were measured, and if so, it would be interesting to show absorption contribution from the nitro-aromatics across the whole spectrum.

The absorption spectra of several nitro-aromatic compounds is known in the literature. It would therefore be possible to show their contribution in the spectrum. However, the wavelength dependence is generally decreasing steadily (Xie et al. 2017) with increasing wavelength, except for 4-nitrocatechol, and would hence show a similar behavior as given for all potential BrC molecules.

5. Figure S3: second panel from top: no color differentiation for the two AAE parameters plotted.

We included a color differentiation.

6. Figure S8. The correlation of gas-phase BrC and temperature is based on exponential fit (y=e(0.15*x)). How is the correlation between temperature and particle phase BrC like? The authors stated "Figure S8 shows that BrC in the gas phase had a good correlation (r=0.4) with temperature." I recommend changing "good" to "moderate".

There is no significant correlation between temperature and particle phase BrC (0.02). Particle phase BrC appears to be dominated by low volatile compounds.

[Figure]

Figure. Correlation between particle phase BrC and temperature.

We changed the sentence on the gas-phase BrC as follows:

Figure S8 shows that BrC in the gas phase had a moderate positive correlation with temperature.

**References:**

Duan, Jing, Ru-Jin Huang, Chunshui Lin, Jincan Shen, Lu Yang, Wei Yuan, Ying Wang, Yi Liu, and Wei Xu.: Aromatic Nitration Enhances Absorption of Biomass Burning Brown Carbon in an Oxidizing Urban Environment, Environ. Sci. Technol., 58, 17344-54, https://doi.org/10.1021/acs.est.4c05558, 2024.

Jiang, F., J. W. Song, J. Bauer, L. Y. Gao, M. Vallon, R. Gebhardt, T. Leisner, S. Norra, and H. Saathoff.: Chromophores and chemical composition of brown carbon characterized at anurban kerbside by excitation-emission spectroscopy and mass spectrometry, Atmos. Chem. Phys., 22, 14971-86, https://doi.org/10.5194/acp-22-14971-2022, 2022.

Lack, D. A., and J. M. Langridge. 2013. 'On the attribution of black and brown carbon light absorption using the Ångström exponent', Atmos. Chem. Phys., 13, 10535-43, https://doi.org/10.5194/acp-13-10535-2013, 2013.

Lin, P., P. K. Aiona, Y. Li, M. Shiraiwa, J. Laskin, S. A. Nizkorodov, and A. Laskin.: Molecular Characterization of Brown Carbon in Biomass Burning Aerosol Particles, Environ. Sci. Technol., 50, 11815-24, https://doi.org/10.1021/acs.est.6b03024, 2016.

Lin, P., L. T. Fleming, S. A. Nizkorodov, J. Laskin, and A. Laskin.: Comprehensive Molecular Characterization of Atmospheric Brown Carbon by High Resolution Mass Spectrometry with Electrospray and Atmospheric Pressure Photoionization, Anal. Chem., 90, 12493-502, https://doi.org/10.1021/acs.analchem.8b02177, 2018.

Mohr, C., Lopez-Hilfiker, F. D., Zotter, P., Prevot, A. S. H., Xu, L., Ng, N. L., Herndon, S. C., Williams, L. R., Franklin, J. P., Zahniser, M. S., Worsnop, D. R., Knighton, W. B., Aiken, A. C., Gorkowski, K. J., Dubey, M. K., Allan, J. D., and Thornton, J. A.: Contribution of Nitrated Phenols to Wood Burning Brown Carbon Light Absorption in Detling, United Kingdom during Winter Time, Environ. Sci. Technol. 47, 6316-6324, https://doi.org/10.1021/es400683v, 2013.

Moschos, V., M. Gysel-Beer, R. L. Modini, J. C. Corbin, D. Massabo, C. Costa, S. G. Danelli, A. Vlachou, K. R. Daellenbach, S. Szidat, P. Prati, A. S. H. Prevot, U. Baltensperger, and I. El Haddad. 2021. : Source-specific light absorption by carbonaceous components in the complex aerosol matrix from yearly filterbased measurements, Atmos. Chem. Phys., 21, 12809–12833, https://doi.org/10.5194/acp-21-12809-2021, 2021.

Tang, J., Li, J., Su, T., Han, Y., Mo, Y. Z., Jiang, H. X., Cui, M., Jiang, B., Chen, Y. J., Tang, J. H., Song, J. Z., Peng, P. A., and Zhang, G.: Molecular compositions and optical properties of dissolved brown carbon in biomass burning, coal combustion, and vehicle emission aerosols illuminated by excitation-emission matrix spectroscopy and Fourier transform ion cyclotron resonance mass spectrometry analysis, Atmos. Chem. Phys., 20, 2513–2532, https://doi.org/10.5194/acp-20-2513-2020, 2020.

Xie, M., Chen, X., Hays, M. D., Lewandowski, M., Offenberg, J., Kleindienst, T. E., and Holder, A. L.: Light Absorption of Secondary Organic Aerosol: Composition and Contribution of Nitroaromatic Compounds, Environ. Sci. Technol., 51, 11607– 11616, https://doi.org/10.1021/acs.est.7b03263, 2017.

Xu, J. Z., Hettiyadura, A. P. S., Liu, Y. M., Zhang, X. H., Kang, S. C., and Laskin, A.: Regional Differences of Chemical Composition and Optical Properties of Aerosols in the Tibetan Plateau, J. Geophys. Res.-Atmos., 125, e2019JD031226, https://doi.org/10.1029/2019jd031226, 2020.

---

## Author Response (AR2)

**Response to reviewers' comments on "Brown carbon aerosol in rural Germany: sources, chemistry, and diurnal variations" (egusphere-2024-1848)**

The authors kindly thank the reviews for the careful review of the manuscript, and the helpful comments and suggestions, which improve the manuscript a lot. All the comments are addressed below point by point, with our responses in blue, and the corresponding revisions to the manuscript in red. All updates of the original manuscript are marked in the revised version.

**Editor**

Please carefully respond to the new referee reports. There are several comments raised by the reviewers that need your attention. It seems both of the reviewers suggest moving this to a Measurement Report style due to the large uncertainties in the presentation of the data. I agree with this suggestion from the reviewers. If you agree as well, we will make this change before publication would occur. In any case, I would like to request that you carefully review their new comments and provide a point-by-point response and track changes version of the manuscript that considers these comments.

We agree to publish this manuscript as a Measurement Report with following title:

Measurement report: Brown carbon aerosol in rural Germany: sources, chemistry, and diurnal variations

We also added dataset DOI links in the manuscript.

"The data related to this article are accessible at KIT open data (https://doi.org/10.35097/d0prpzkxqkq2t09y, Jiang et al., 2024).

**Reviewer #1**

The authors have largely addressed the points raised in my original review. Thank you. I have a few additional minor points to raise:

1. Provide units for the new "sensitivity factors".

We changed the sensitivity factors into sensitivity with units of cps ppt$^{-1}$ throughout the manuscript and supplement.

2. Provide the wavelength in Table S5 for the MAC value.

We added the wavelength as given below:

Nitro aromatic compounds in particle phase detected at KIT Campus Nord, including chemical formula, mass absorption coefficient (MAC) at 365 nm, concentration range, and average concentration (mean ± standard deviation).

3. Double check the new text in the paper, as there are a couple of typos and wording problems in it.

We double checked the manuscript for typos and wording.

4. Add the dates at which the calibrations were done. If the calibrations were not close in time to the field campaign, then a comment indicating the associated uncertainty should be included.

First calibrations for nitro aromatic compounds (NACs) were done directly after the field campaign. However, due to technical problems during the first calibration, it was repeated in August 2024. We added this information as follows:

"First calibrations for nitro aromatic compounds (NACs) were done directly after field campaign. However, due to technical problems, the calibration of 4-nitrophenol, 4-nitrocatechol, 2-methyl-4-nitropehnol, and 4-methyl-5-nitrocatechol was repeated in August 2024. These results are shown in Figure S9. Despite the large time between measurements and second calibration, we have indications from repeated measurements of formic acid that the sensitivity of the instrument didn't change substantially over this time period. Please note that this leads to an additional uncertainty of about 20%."

5. Overall, this paper provides coupled absorption and mass spectrometry measurements related to BrC in Germany. There is merit to publication because the measurements are new. The absorption measurements are fine, especially now that the aethalometry approach has been better described. However, I still feel that this work is "semi-quantitative" with respect to BrC molecule quantification. For example, I feel it is not appropriate to have statements such as the following in the Abstract: "The 178 potential BrC molecules only accounted for $2.6 \pm 1.5\%$ of the total organic mass, but can explain $14 \pm 13\%$ of the total BrC absorption at 370 nm" In particular, by calibrating only a few nitrophenols, I do not believe that one can claim to make a statement about closure between BrC absorption and the species giving rise to that absorption. Moreover, the organic aerosol mass was not measured.

We agree that there are large uncertainties from CIMS sensitivity, mass absorption cross-section for nitro-aromatics, and estimated organic matter concentrations. Therefore, we deleted the absorption results and discussions from potential BrC molecules in section 3.3. In addition, we deleted this statement of "The 178 potential BrC molecules only accounted for $2.6 \pm 1.5\%$ of the total organic mass, but can explain $14 \pm 13\%$ of the total BrC absorption at 370 nm" in abstract and conclusion.

Since we calibrated the NACs sensitivity of CIMS and known the mass absorption cross-section of NACs (Xie et al., 2017). We calculated the absorption contribution of seven NACs for total BrC absorption, as shown in new Figure 3.

We add a new statement in the abstract:

"The average light absorption of seven NACs in the particle phase was $0.2 \pm 0.2$ Mm$^{-1}$, contributing to $2.2 \pm 2.1\%$ of total BrC absorption at 370 nm."

We add new statements to the results and discussion section 3.3:

"We calculated the average light absorption of seven nitro aromatic compounds (NACs) by using the mass absorption coefficients (MAC$_{365}$, Xie et al., 2017), given in Table S5 and the average concentrations measured. Based on this, the mean light absorption of the sum of the seven NACs was calculated to be $0.2 \pm 0.2$ Mm$^{-1}$. The absorption of the seven NACs contributed to $2.2 \pm 2.1\%$ of total BrC absorption at 370 nm (Figure 3b)."

[Figure]

Figure 3. (a) A stacked plot showing the main contributions to aerosol absorption from brown carbon and black carbon based on the seven wavelengths measured by the aethalometer AE33. The contribution of seven NACs to the total aerosol absorption is indicated in red at 370 nm. (b) Average absorption contribution of seven NACs to total absorption by BrC. The red: seven NACs; the gray: unidentified-BrC molecules.

6. Also, I am still puzzled by the following statement in the paper: "Please note that the sensitivity of CIMS for different organic compounds varies by a few orders of magnitude. Sensitivity uncertainties were taken into account in the calculation of the overall uncertainties of CIMS concentrations (±60%) …" If CIMS sensitivities can vary by orders of magnitude (which I agree with), then how can the overall uncertainties in the CIMS concentrations be ±60%? This doesn't make sense.

We deleted this sentence.

7. My advice would be to remove such statements about absorption closure (such as that in the Abstract) from the paper, because I don't believe they are quantitatively justified.

We deleted this statement of "The 178 potential BrC molecules only accounted for $2.6 \pm 1.5\%$ of the total organic mass, but can explain $14 \pm 13\%$ of the total BrC absorption at 370 nm" in abstract, results, and conclusion.

**References:**

Xie, M., Chen, X., Hays, M. D., Lewandowski, M., Offenberg, J., Kleindienst, T. E., and Holder, A. L.: Light Absorption of Secondary Organic Aerosol: Composition and Contribution of Nitroaromatic Compounds, Environ. Sci. Technol., 51, 11607– 11616, https://doi.org/10.1021/acs.est.7b03263, 2017.

Jiang, F., Saathoff, H., Ezenobi, U., Song, J., Zhang, H., Gao, L., and Leisner, T.: Dataset for the publication: Brown carbon aerosol in rural Germany: sources, chemistry, and diurnal variations, Karlsruhe Institute of Technology [data set], https://doi.org/10.35097/d0prpzkxqkq2t09y, 2024.

**Reviewer #2**

**General comments**

1. After reviewing the revised manuscript and author's response, I am not convinced by the quantitative conclusions drawn from this study based on numerous assumptions. The study used too many empirical or averaged parameters (e.g. CIMS sensitivity, mass absorption cross-section for nitro-aromatics, organic matter concentrations) that could lead to a very large error bar on any of the quantities derived from the analysis. In fact, the mass concentrations of the identified brown carbon compounds and their contribution to organic mass and brown carbon absorption changed significantly in the revised manuscript without proper explanation.

We agree that we used to many assumptions to estimate the potential absorption of the 178 potential BrC molecules. Therefore, we deleted the absorption results and discussions from potential BrC molecules in section 3.3. In addition, we deleted this statement of "The 178 potential BrC molecules only accounted for $2.6 \pm 1.5\%$ of the total organic mass, but can explain $14 \pm 13\%$ of the total BrC absorption at 370 nm" in abstract and conclusion.

In the revised manuscript we use mass concentrations of the nitro aromatic compounds based on a calibration of our CIMS instead of an estimated average sensitivity. Together with literature values (Xie et al., 2017) of their mass absorption cross sections, we can quantify their contribution to BrC absorption, as show in new Figure 3.

We add a new statement in the abstract and conclusion:

"The average light absorption of seven NACs in the particle phase was $0.2 \pm 0.2$ Mm$^{-1}$, contributing to $2.2 \pm 2.1\%$ of total BrC absorption at 370 nm."

We add new statements to the results and discussion:

"We calculated the average light absorption of seven nitro aromatic compounds by using the mass absorption coefficients (MAC$_{365}$, Xie et al., 2017), given in Table S5 and the average concentrations measured. Based on this, the mean light absorption of the sum of the seven NACs was calculated to be $0.2 \pm 0.2$ Mm$^{-1}$. The absorption of the seven NACs contributed to $2.2 \pm 2.1\%$ of total BrC absorption at 370 nm (Figure 3b)."

[Figure]

Figure 3. (a) A stacked plot showing the main contributions to aerosol absorption from brown carbon and black carbon based on the seven wavelengths measured by the aethalometer AE33. The contribution of seven NACs to the total aerosol absorption is indicated in red at 370 nm. (b) Average absorption contribution of seven NACs to total absorption by BrC. The red: seven NACs; the gray: unidentified- BrC molecules.

**References:**

Xie, M., Chen, X., Hays, M. D., Lewandowski, M., Offenberg, J., Kleindienst, T. E., and Holder, A. L.: Light Absorption of Secondary Organic Aerosol: Composition and Contribution of Nitroaromatic Compounds, Environ. Sci. Technol., 51, 11607– 11616, https://doi.org/10.1021/acs.est.7b03263, 2017.